# Evaluation of Index-Based Methods for Impervious Surface Mapping from Landsat-8 to Cities in Dry Climates; A Case Study of Buraydah City, KSA

Hussein Almohamad * and Ibrahim Obaid Alshwesh

Department of Geography, College of Arabic Language and Social Studies, Qassim University, Buraydah 51452, Saudi Arabia; shoiesh@qu.edu.sa
* Correspondence: h.almohamad@qu.edu.sa

**Abstract:** The natural landscape is fast turning into impervious surfaces with the increase in urban density and the spatial extent of urbanized areas. Remote sensing data are crucial for mapping impervious surface area (ISA), and several methods for ISA extraction have been developed and implemented successfully. However, the heterogeneity of the ISA spectra and the high similarity of the ISA spectra to those of bare soil in dry climates were not adequately addressed. The objective of this study is to determine which spectral impervious surface index best represents impervious surfaces in arid climates using two seasonal Landsat-8 images. We attempted to compare the performance of various impervious surface spectral Index for ISA extraction in dry climates using two seasonal Landsat-8 data. Specifically, nine indices, i.e., band ratio for the built-up area (BRBA), built-up area extraction method (BAEM), visible red near infrared built-up index (VrNIR-BI), normalized ratio urban index (NRUI), enhanced normalized difference impervious surfaces index (ENDISI), dry built-up index (DBI), built-up land features extraction index (BLFEI), perpendicular impervious surface index (PISI), combinational biophysical composition index (CBCI), and two impervious surface binary methods (manual method and ISODATA unsupervised classification). According to the results, PISI and CBCI combined with the manual method had the best accuracy with 88.5% and 88.5% overall accuracy (OA) and 0.76 and 0.81 kappa coefficients, respectively, while DBI combined with the manual method had the lowest accuracy with 75.37% OA and 0.56 kappa coefficients. PISI is comparatively more stable than the other approaches in terms of seasonal sensitivity. The ability of PISI to discriminate ISA from soil and vegetation accounts for much of its good performance. In addition, spring is the ideal time of the year for mapping ISA from Landsat-8 images because the impervious surface is generally less likely to be confused with bare soil and sand at this time of year. Therefore, this study can be used to determine spectral indices for studying ISA extraction in drylands in conjunction with binary approaches and seasonal effects.

**Keywords:** urban area; index-based method; image analysis; Landsat-8; dry climates

## 1. Introduction

Studying the temporal and spatial evolution of cities can serve as an indicator of urban development and environmental quality [1–4]. The expansion of impervious surfaces destroys ecosystems and affects the migration of organisms [5]. Therefore, precise measurements of urban impervious surface area distributions and impervious surface coverage are of great significance for environmental protection studies [6], as well as natural resource management to ensure food, water, and energy supplies. Data from remote sensing platforms provide up-to-date information and a comprehensive overview of landscape features and changes in urban areas [7,8]. Mapping such an urban impervious built-up area using remote sensing is not only cost-effective but also time-saving [9]. Urban monitoring and planning can help minimize the degradation of the natural environment and mitigate

climate change. In recent decades, various remote sensing methods, such as the Landsat and Sentinel missions, have become widely available as a data source for mapping and monitoring land use and land cover [10,11]. Owing to rapid urbanization, land cover in many urban areas of the world is changing faster than ever before [12,13]. However, accurately extracting urban areas is a difficult task because several factors influence the classification. Urban areas consist of very different materials, both spatially and spectrally, such as concrete, asphalt, metal, plastic, and glass, whose spectra can be similar to those of other materials, such as bare land, wasteland, and sand.

In recent years, different methodologies have been proposed to extract impervious surfaces and urban extent from satellite data, whose results can vary depending on the imagery used and the study areas. Multiple regression [14], artificial neural networks [15,16], spectral unmixing [17,18], object-oriented and knowledge-based classification methods [19–22], and the use of development indices are examples of these methods, which, due to differences in geographic locations, spatial scales, landform patterns, etc. Have advantages and disadvantages.

Separating bare soil, alluvial, and fluvial sediment regions from urban areas is a fundamental and difficult task to obtain accurate estimates for urban regions. Consequently, urban development rates and development trends may be misjudged at the very first step of urban planning. Once barren land is efficiently distinguished from developed land, this improves the assessment of urban expansion and contributes to other developments, such as land management [10]. Methods based on ISA indices have proven to be effective techniques for extracting cities due to their simplicity, ease of implementation, and speed. So far, several build-up indices have been proposed in the literature that is suitable for medium to coarse spatial resolution data, such as Landsat imagery. However, their performance depends on spatial resolution, acquisition time, and surface relief [1,23–27]. An index-based ISA extraction approach for multispectral imaging typically finds the brightest and faintest reflectance bands of the impervious surface and enhances the intensity contrast between the impervious surface and the background through mathematical operations on selected bands [26].

Numerous spectral indices generated by Landsat and Sentinel-2 have been proposed by researchers around the world to extricate ISA from other fields. Typical indices include (but are not limited to) the normalized difference built-up index (NDBI) [28], Index-based built-up Index (IBI) [2], Band ratio for built-up area (BRBA) [24], normalized difference bareness index (NDBaI) [29], built-up area extraction method (BAEM), a built-up and bare land index (BBI$_{OLI}$) [2], visible red near infrared built-up index (VrNIR-BI) [30], normalized ratio urban index (NRUI) [31], enhanced normalized difference impervious Surfaces index (ENDISI, dry built-up index (DBI) [4] to map built-up areas from Landsat 8, combinational build-up index (CBI) [32], built-up land features extraction index (BLFEI) [33], modified normalized difference impervious surface index (MNDISI), perpendicular impervious surface index (PISI) [34], modified normalized urban areas composite index (MNUACI) [35] have been employed in various studies.

A number of studies have compared the performance of ISA indicators in many regions under different climate types, for example. As-syakur et al. [1] proposed the enhanced built-up and Bareness Index (EBBI) index and compared it with previous indices, such as IBI, NDBI, and UI, in humid tropical climate regions. They concluded that EBBI is a more effective index than other indices when applied to Landsat ETM+. Bhatti and Tripathi [3] (2014) proposed the BAEM using Landsat 8 data in semi-arid climate regions and compared it with modified NDBI for Landsat 8 data. The study resulted in an increased output of the BAEM approach. Bouzekri et al. [36] developed the built-up area extraction index (BAEI) and applied it in the Mediterranean climate region. When compared with the BRBA, NDBI, normalized built-up area index (NBAI), and new built-up index (NBI), the new index provided higher accuracy when using Landsat 8 data. Tian et al. (2018) [34] proposed the PISI using Landsat 8 data and compared it with BCI and NDBI. The findings show that in diverse climate regions, from humid to semi-arid, PISI is more precise and has

greater separability for ISA and soil, as well as ISA and vegetation in the ISA extraction than the BCI and NDBI. Li et al. [37] attempted to assess the performance of different spectral indices (PISI, NBAI, BCI, and BCI) for ISA extraction using multi-seasonal Sentinel-2 images in a humid subtropical climate. Results showed that PISI combined with the ISODATA classification method achieved the highest accuracy of 92.6%.

A review of the literature on spectral indices leads us to conclude that the performance of these indicators depends on spatial resolution, type of climate, acquisition time, surface relief, and the materials of which they are composed [35]. In areas where vegetation is scarce, it is difficult to distinguish impervious surfaces from bare soil. An evaluation of the performance and accuracy of these techniques has been made in cities with humid to semi-arid climates, but these indices have not been applied in arid climates, where about one-third of the world's population lives and where it is increasingly difficult to separate the barren soils and wadi sediments, sand dunes, and salt marshes (sebkha) from urban areas. Therefore, it is necessary to systematically and thoroughly analyze the efficiency of the impervious surface index method using Landsat-8 imagery and its applicability in dry climate cities where urbanization is expected to increase mainly due to increasing population growth in Africa and Asia, which have more vulnerable environmental systems. The first objective of this work is to evaluate the performance of the results of the index-based methods proposed, such as VrNIR-BI, CBCI, ENDISI, PISI, BRBI, BLFEI, DBI, BCI, and RNUI, using Landsat-8 imagery in arid climates, and to investigate the impact of seasonal variations on the ISA extracted to check the consistency of the results. The second objective is to evaluate the performance of the ISODATA classification method and manual threshold for separating impervious surfaces from index images. It is hypothesized that the spectral indicators studied differ in their performance in dry environments since some of these indicators were tested in humid environments and others in semi-humid and semi-dry environments. This hypothesis will be tested for validity within the analysis.

The organization of the document is as follows: Section 2 highlights the selection of the study area and the data collection procedure and describes the methodology, including spectral indices. Sections 3 and 4 present the results of comparative analysis of the separability, thresholding, and extraction precision analysis of spectral indices and discuss their application to areas with different artificial surface features of the city. Finally, in conclusion, Section 5 summarizes the main findings of this study and presents the proposed topics for future research

## 2. Materials and Methods

### 2.1. Study Area

The study is conducted in Buraydah City, the capital of Qassim province and one of the largest megacities in Central Saudi Arabia (Figure 1). Buraydah is located at 26°21′33.23″ N and 43°58′54.52″ E, and the geographical area of Buraydah city is approximately 1300 km$^2$. The city is located at an elevation of approximately 610 m Above Sea Level (A.S.L). Most of the city has flat terrain decorated with sand hills, sand seas, and agricultural land.

Buraydah City is dominated by a hot desert climate (BWh) according to the Koppen climate classification, characterized by heat and dryness in the summer and moderately cold temperatures in the winter [38]. The average air temperature of the city ranges from 13 °C in winter to 36 °C in summer. Annual precipitation totals 72 mm, mainly from November to May. The city, which is one of the largest developing cities in Saudi Arabia, has experienced significant population growth in recent decades, rising from 7377 in 1960 to 700,000 in 2022 [39]. Buraydah is currently experiencing the largest expansion in its history. On the one hand, rapid growth provides numerous benefits to its citizens, such as job opportunities, convenient transportation, beautiful views, and a modern living environment. On the other hand, it causes significant air pollution, water pollution, and agricultural land loss in the city.

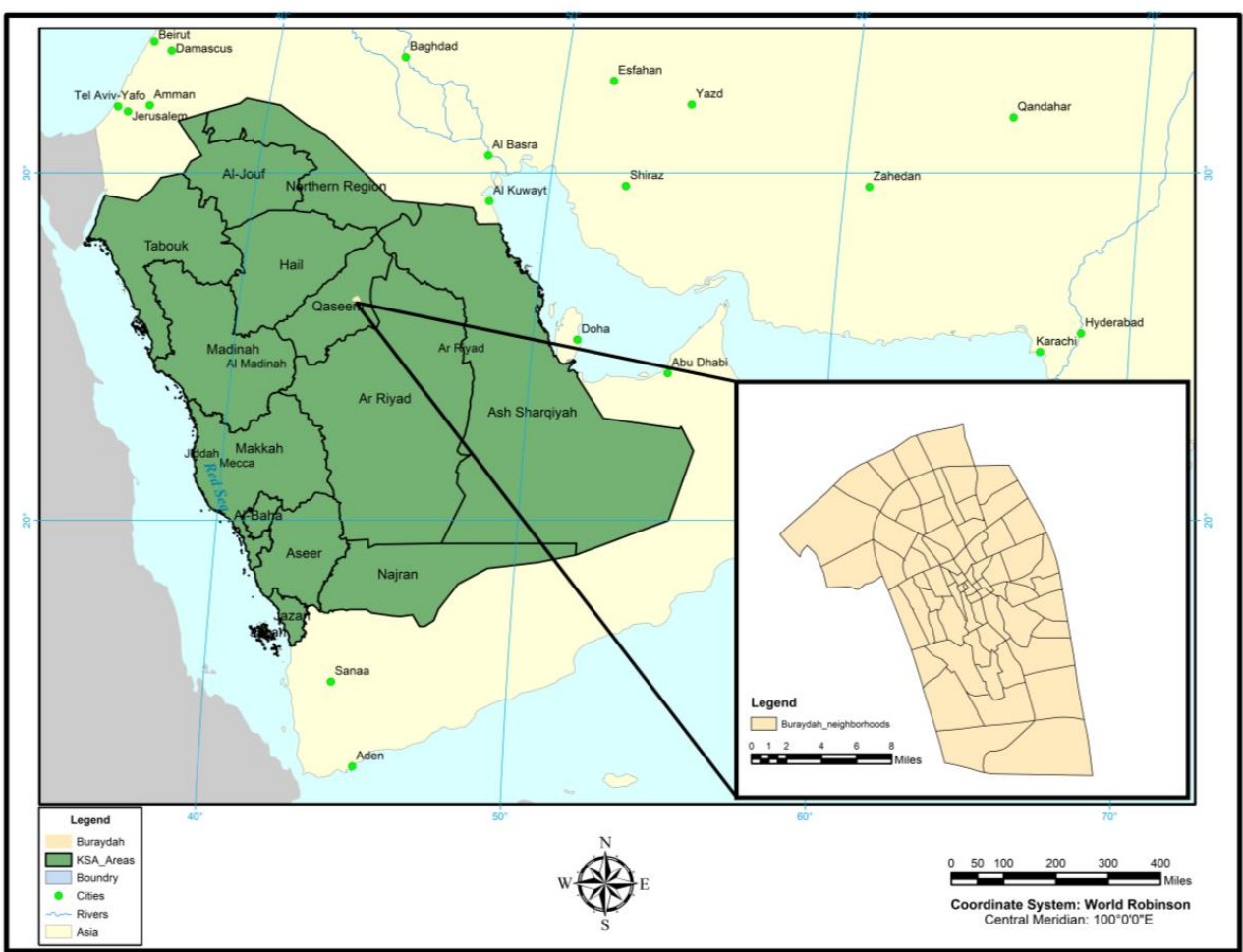

**Figure 1.** Study area.

*2.2. Method*

2.2.1. Data

For this study, Landsat 8 (OLI & TIRS) images were acquired from August 2020 (dry season) and March 2021 (relative wet season) on path 168 and row 042 (cloud cover = 0.01%) to extract the ISA area. Landsat 8 images acquired on two different dates were used in order to check the reliability of the results as well. Landsat 8 (OLI & TIRS) has eight reflectance bands with a resolution of 30 m, one panchromatic band with a resolution of 15 m, and two thermal bands with a resolution of 100 m. The specifications of Landsat 8 (OLI & TIRS) are given in Table 1. The raw data were analyzed using ArcGIS 10.6 software with atmospheric correction. The images are in UTM projection (zone 38 N) and were retrieved from the United States Geological Survey (USGS) website, Earth Explorer. Figure 2 shows the method of data processing and analysis.

**Table 1.** Landsat-8 data for two seasons acquired for ISA Indices.

| Landsat Scene Identifier (Path/Row: 168/042) | Sensing Date | Sensor Identifier | Cloud (%) | Season |
|---|---|---|---|---|
| LC81680422020243LGN00 | 2020/08/30 | OLI_TIRS | 0.0 | Summer |
| LC81660432021071LGN00 | 2021/03/12 | OLI_TIRS | 0.0 | Spring |

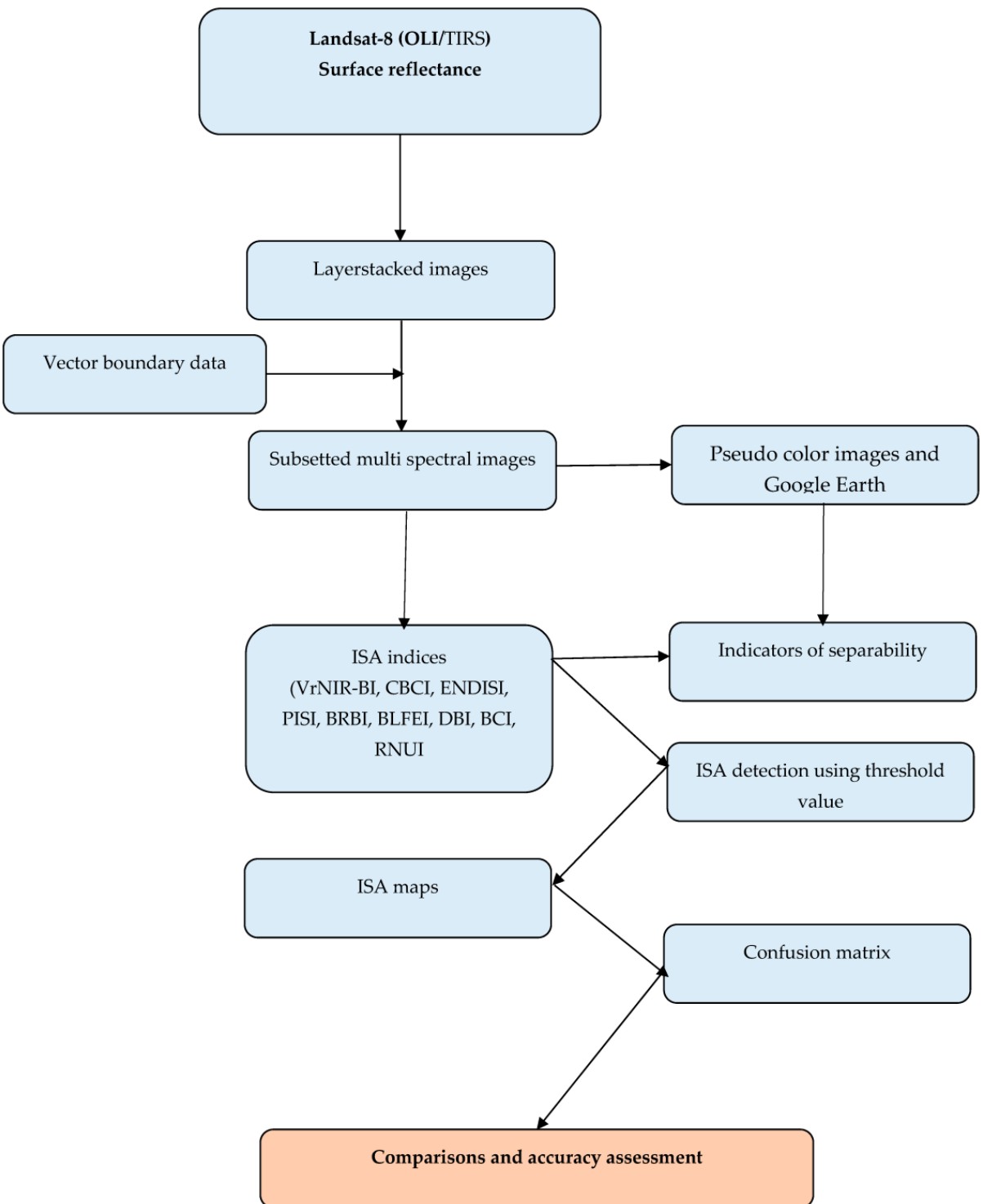

**Figure 2.** Flowchart of the methodology (data collection, ISA detection and validation).

Impervious Surface Indices

One question remains unresolved, although several lines of evidence for the extraction of impervious surfaces have recently been proposed in the literature. We do not yet know the common spectral features of impervious surfaces that function globally under different climate types. It also appears that the indicators have not been tested in very dry climates, as is the case in our study area. Therefore, in this study, we will use nine indicators of impervious surfaces that have been used in more than one climate type or that have been

used in an area of semi-arid climate that most closely resembles the climate of our study area (dry climates).

- The biophysical composition index (BCI): the BCI was proposed by Deng and Wu [40] to identify different urban biophysical compositions based on the Tasseled Cap (TC) transform, a transformation able to compress spectral information from multiple bands into fewer space scenes, highlighting spectral characteristics of different land cover types [3]. It is calculated as:

$$BCI = \frac{(TC1 + TC3)/2 - TC2}{(TC1 - TC3)/2 + TC2} \tag{1}$$

where TC1 is associated with high albedo materials (brightness), TC2 with vegetation (greenness), and TC3 with low albedo (wetness). Hence, the index is computed as

The proposed index was applied to images registered by different satellites over different regions of the world.

- Built-up land features extraction index (BLFEI): Bouhennache et al. [33] developed a built-up land features extraction index (BLFEI) using green, red, SWIR1, and SWIR2 operational land imager (OLI) bands. Because asphalt-like roads and arid soil have nearly similar spectral responses for the spectrum from NIR to SWIR1, the NIR spectral channel was not utilized. In BLFEI, vegetation had the lowest value, while water had the greatest. The values of impervious surface areas were lower than water and higher than bare soil areas. It is calculated as:

$$BLFEI = \frac{((Green + Red + SWIR2)/3) - SWIR1}{((Green + Red + SWIR2)/3) + SWIR1} \tag{2}$$

- Perpendicular Impervious Surface Index (PISI): Tian et al. (2018) [34] proposed a reference line equation known as the PISI, which distinguishes ISA from bare soil, using feature spaces in the blue and NIR bands. Though it uses only two bands, it has higher accuracy in separating the impervious area from the bare soil area up to this point in time. PISI performed significantly better than the BCI and NDBI indices. It applies to most optical sensors due to the use of only blue and NIR bands. The example of PISI can be replicated in numerous other RS applications. Similar to BBI, PISI increases the separability between ISA and bare soil and between ISA and vegetation areas. The formula for PISI can be expressed as follows:

$$PISI = 0.8192 \, Blue - 0.5735 \, NIR + 0.075 \tag{3}$$

where Blue and NIR denote the reflectance of the Blue and NIR bands, respectively.

- City Biophysical Component Index (CBCI): The modified bare soil index (MBSI) and optimized soil adjusted vegetation index (OSAVI) were merged by Zhang et al. [27] to create the city biophysical component index (CBCI), which effectively shows the four major biophysical components of cities. The following are the formulas:

$$CBCI = (A + 1)*MBSI - OSAVI + A \tag{4}$$

$$CBCI = \frac{(Red - Green)*2}{(Red + Green - 2)} \tag{5}$$

$$OSAVI = \frac{NIR - Red}{NIR + Red + 0.16} \tag{6}$$

where green, red and NIR are the surface reflectance values of the red, green, and near-infrared, respectively. A is a correction factor, which was set to 0.5 in this study to distinguish between impervious surfaces, bare land, vegetation, and water.

- Band ratio for built-up area (BRBA): Waqar et al. introduced two new BRBA [24] that were employed to extract the built-up areas of Islamabad city, Pakistan. Here, BRBA used red and only one SWIR band. The study claimed to increase built-up extraction accuracy by 10–13% compared to NDBI and NBI. BRBA is computed according to the expression:

$$\text{BRBA} = \frac{\text{Red}}{\text{SWIR1}} \tag{7}$$

- Dry buildup index (DBI): Rasul et al. [4] suggested a DBI based on Landsat OLI blue and thermal bands. The DBI assumes that built-up regions have less vegetation and, therefore, low NDVI values. Therefore, a reduction in NDVI can further improve the characteristics of built-up areas. DBI has an overall classification accuracy for urban areas of 93% for an arid climate, as is the case in the city of Erbil. DBI is not suitable in densely forested urban areas. There are several limitations to the thermal data that must be considered before use: The spectral difference in thermal bands is small and often shows phenological and diurnal differences between urban and non-urban areas.

$$\text{DBI} = \frac{\text{Blue} - \text{TIR1}}{\text{Blue} + \text{TIR1}} - \text{NDVI} \tag{8}$$

- Enhanced Normalized Difference Impervious Surfaces Index (ENDISI): Chen et al. [41] developed the enhanced normalized difference impervious surface index (ENDISI) to amplify the difference between impervious surfaces (ISs) and pervious surfaces (PSs), where preprocessing, such as removing waterbodies, is not required. The SWIR-1 band to SWIR-2 band ratio, the MNDWI, and the blue band are chosen by ENDISI as enhancement and inhibitor factors, respectively. The equation is how ENDISI is written (9):

$$\text{ENDISI} = \frac{\text{Blue} - a\left(\frac{\text{SWIR1}}{\text{SWIR2}} + \text{MNSWI}^2\right)}{\text{Blue} + a\left(\frac{\text{SWIR1}}{\text{SWIR2}} + \text{MNSWI}^2\right)} \tag{9}$$

where

$$a = \frac{2\text{Blue}_{\text{mean}}}{\left(\frac{\text{SWIR1}}{\text{SWIR2}}\right)_{\text{mean}} + \text{MNDWI}^2_{\text{mean}}} \tag{10}$$

where Blue, SWIR1, and SWIR2 are the surface reflectance values of the blue, first shortwave-infrared, and second shortwave-infrared bands, respectively. Mean is the mean image value. a is a correction factor calculated by Equation (10), which is used to stretch the value of ENDISI to $-1\sim1$.

- Normalized Ratio Urban Index (NRUI) Piyoosh and Ghosh [31] have proposed the NRUI, and they have mentioned that using the panchromatic (PAN) band (Band 8) of Landsat 8 data leads to an overall improvement in discriminating between ISA, bare soil, and vegetation. Subsequently, the MNDSI was utilized to develop a new normalized ratio urban index (NRUI) by improving the capability of the BCI in two phases. First, a ratio urban index (RUI) was developed, which discriminates urban and soil better than the BCI. Second, the RUI was further improved, which then became known as NRUI and is able to distinguish urban areas from the ground even better than the RUI.

$$\text{RNUI} = \frac{\text{RUI} - \text{MNDSI}}{\text{RUI} - \text{MNDSI}} \tag{11}$$

$$\text{MNDSI} = \frac{\text{SWIR2} - \text{PAN}}{\text{SWIR2} + \text{PAN}} \tag{12}$$

$$\text{RUI}\frac{\text{BCI}}{\text{MNDSI}} \tag{13}$$

Visible red near infrared built-up index (VrNIR-BI): Estoque and Murayama [30] (2015) proposed the visible green-based built-up index (VgNIR-BI) and VrNIR-BI, which is a simple and accurate index and is applied for Landsat 7, as well as Landsat 8. Among these two, VrNIR-BI works better than the VgNIR-BI index, and these are best at classifying ISA and dry vegetated areas. For this reason, the VrNIR-BI was used in this study, and the expression of this index is given by:

$$\text{VrNIR} - \text{BI} = \frac{\text{Red} - \text{NIR}}{\text{Red} + \text{NIR}} \tag{14}$$

where red and NIR are the surface reflectance values of the red and near-infrared, respectively [42].

### 2.2.2. Separability Analysis

The purpose of the separability analysis is to see how well the three indices distinguish between the three components (ISA, soil, and vegetation). To obtain assess the separability (accurate proportions) of ISA, bare soil, and vegetation pixels from Landsat-8 imagery, we randomly selected three main land cover types by interpreting high-resolution imagery (Worldview-3 imagery) using imagery from 2020. We examined the land cover class separability between urban areas and other classes using the spectral discrimination index (SDI) [43,44], Jeffries–Matusita (J–M) distance [45], and transformed divergence (TD) [46], which are representative separability measurements. The SDI (question), (J–M) distance (question) and TD (question) are represented as follows:

The formulas for SDI, J–M distance, and TD are as follows:

$$JM_{ij} = 2\left(1 - exp^{B_{ij}}\right) \tag{15}$$

$$B_{ij} = \frac{1}{8}\left(u_i - u_j\right)^2 \frac{2}{\left(v_i^2 - v_{ij}^2\right)} + \frac{1}{2}\ln\left(\frac{v_i^2 + v_j^2}{2v_i u v_j}\right) \tag{16}$$

where $B_{ij}$ is the Bhattacharyya distance, $u_i$ and $u_j$ are the means, and $v_i$ and $v_j$ are the variance of adjacent segments $i$ and $j$, respectively.

The following degrees of distinguishability of pairs of land cover classes were adopted in the research: low separability: from 0 to 0.999, moderate separability: from 1 to 1.299, and very good separability: from 1.3 to 1.414.

$$SDI = \frac{\left|u_i - u_j\right|}{\sigma_i + \sigma_i} \tag{17}$$

$$TD = 2\left[1 - \exp\left(-\frac{D}{8}\right)\right] \tag{18}$$

$$D = \frac{1}{2}tr\left[(C_i - C_j)(C_i^{-1} - C_{ij}^{-1})\right] + \frac{1}{2}tr\left[(C_i^{-1} - C_{ij}^{-1})(u_i - u_j)(u_i - u_j)^T\right] \tag{19}$$

where $C$ is the covariance matrix of the class and tr is the trace of the matrix.

PISI, BCI, and NDBI were then applied to grayscale photos of the four subregions.

These three indicators were used to evaluate the urban zone indicators and were classified into three categories, as shown in Table 2. Specifically, the J–M distance indicates the separability between two classes, with a value less than 1.00 indicating that the two classes are poorly separable. Further, if the J–M distance is larger than 1.3, it indicates a high degree of separability, and if its value is between 1.00 and 1.3, it means the two classes are

moderately separable [47]. Similarly, a TD value of more than 1900 indicates a high degree of separation between the two classes. A value of TD between 1700 and 1900 indicates moderate separation. When it is less than 1700, the two classes overlap with each other, which indicates poor separation [48]. If SDI < 1 means that the classes overlap and the ability to discriminate the classes is poor [33]. When the SDI is greater than 0.99 and less than 1.99, the means are relatively good separated (moderate separability), and the regions are generally easy to distinguish. When the value SDI > 1.99 is very good discrimination of land features, and there is no more overlap [33].

**Table 2.** Class of separability of J–M distance, TD and SDI.

| Class of Separability | J–M | TD | SDI |
|---|---|---|---|
| very good separability | 1.3–1.414 | 1900< | 2< |
| moderate separability | 1–1.299 | 1700–1900 | 1–1.99 |
| low separability | <0.999 | <1700 | 0.99 |

Accuracies Assessment

ISA Classification Using Thresholding

In the extraction accuracy analysis, the threshold is an important factor for land cover extraction using the index-based method. Typically, researchers use their experience and trial-and-error techniques to define a more accurate threshold for a given study that works well for the chosen study time, location, and datasets. However, the threshold often varies spatially and temporally. Therefore, an automatic threshold can speed up the segmentation process. In urban ISA extraction, the exact threshold is crucial [49], such as in Otsu's method, but it works better for two well-distinguishable classes. Therefore, it is not recommended when classifying into multiple classes.

Otsu's method automatically performs histogram-based image thresholding or reduces a grayscale image to a binary image. Otsu's method is usually used to set and determine the optimal thresholds for ISA indices, but Li et al. [37] found it to be the worst method to determine the optimal thresholds for ISA compared to ISODATA and manual thresholding, which were similar and outperformed Otsu's method. Therefore, in this study, manual and ISODATA thresholding algorithms were used to set and determine the optimal thresholds for ISA indices based on classification. The ISODATA algorithm is an iterative process that classifies data elements into different classes using Euclidean distance as a similarity measure. Using high-resolution Google Earth imagery, the resulting clusters were divided into impervious and pervious classes after categorization.

Accuracies Assessment

The satellite imagery in Google Earth of Buraydah, a metropolis in Saudi Arabia, was used to assess accuracy. Random sampling was used to collect sample pixels to compare the accuracy of the different indices and evaluate the difference between them. The accuracy of mapping was expressed in terms of User Accuracy (UA), Producer Accuracy (PA), Overall Accuracy (OA), and Kappa coefficient (κ). The OA was calculated as the ratio between the total number of correctly labeled samples and the total number of tested samples to evaluate the efficiency of the applied algorithms. The PA was used to calculate the probability that the reference sample on the map would be correctly classified. The UA was used as an indicator of the probability that a classified pixel on the urban area classification map would accurately represent that category on the ground. The (k) was used to measure the agreement between the urban area model (indices values) and the actual class values as if it happened by chance.

## 3. Results

### 3.1. Separability Analysis

The index images were uniformly rescaled to a range of values from −1 to 1 to allow comparison between indices. The histogram overlap method, SDI, (J–M) distance, and (TD) were used to evaluate the impervious surface indices mentioned in Section 2. This analysis will help us quantify, at a given point, the degree of overlap of pixel values belonging to each class, which is indeed a general problem for most indices. The land use categories of impervious surface, bare surface, and vegetation are used as spectral signatures in the region of interest (ROI). The impervious surface indices calculated for summer and spring are shown in Figures 3 and 4. Histograms of land features show well-separated classes, such as ISA and vegetation but poor isolation or overlap with barren land for most ISA indices. In Figures 3 and 4, the bare land histograms for BCI, DBI, and ENDISI had a medium wave peak on the right that overlapped with the ISA, while we find that the PISI and CBCI overlap less between the bare land and ISA.

Table 3 shows that all indicators were able to discriminate moderately to well between impervious surfaces and vegetation cover in spring and summer, with a performance preference for spring. The chlorophyll content is high in spring, making it easier to separate impervious surfaces from vegetation cover.

**Table 3.** Separability measures between ISA and soil and ISA and vegetation with different indices.

| | Summer | | | Spring | | |
|---|---|---|---|---|---|---|
| **ISA and Barre Soil** | **J–M** | **TD** | **SDI** | **J–M** | **TD** | **SDI** |
| BLFAI | 1092 | 1225 | 1.30 | 1095 | 1262 | 1.31 |
| BCI | 607 | 498 | 0.44 | 706 | 687 | 0.61 |
| BRBI | 1054 | 1118 | 1.28 | 1003 | 1114 | 1.18 |
| CBCI | 1075 | 1290 | 1.31 | 1073 | 1510 | 1.28 |
| DBI | 753 | 582 | 0.81 | 841 | 708 | 0.94 |
| ENDISI | 508 | 316 | 0.37 | 642 | 505 | 0.59 |
| VegNIR | 1053 | 1112 | 1.27 | 1098 | 1258 | 1.36 |
| NRUI | 945 | 1264 | 0.98 | 1004 | 1384 | 1.04 |
| PISI | 1094 | 1309 | 1.30 | 1078 | 1480 | 1.30 |
| | **Summer** | | | **Spring** | | |
| **ISA and Vegetation** | **J–M** | **TD** | **SDI** | **J–M** | **TD** | **SDI** |
| BLFAI | 1278 | 1635 | 1.84 | 1312 | 1746 | 1.99 |
| BCI | 1231 | 1577 | 1.69 | 1318 | 1760 | 2.02 |
| BRBI | 1254 | 1594 | 1.76 | 1313 | 1726 | 1.99 |
| CBCI | 1293 | 1673 | 1.90 | 1358 | 1933 | 2.31 |
| DBI | 1289 | 1989 | 1.95 | 1348 | 1999 | 2.33 |
| ENDISI | 1186 | 1453 | 1.56 | 1257 | 1602 | 1.70 |
| VrNIR-BI | 1364 | 1914 | 2.33 | 1387 | 1983 | 2.64 |
| NRUI | 1246 | 1978 | 1.37 | 1155 | 1970 | 1.34 |
| PISI | 1311 | 1729 | 2.13 | 1338 | 1806 | 2.13 |

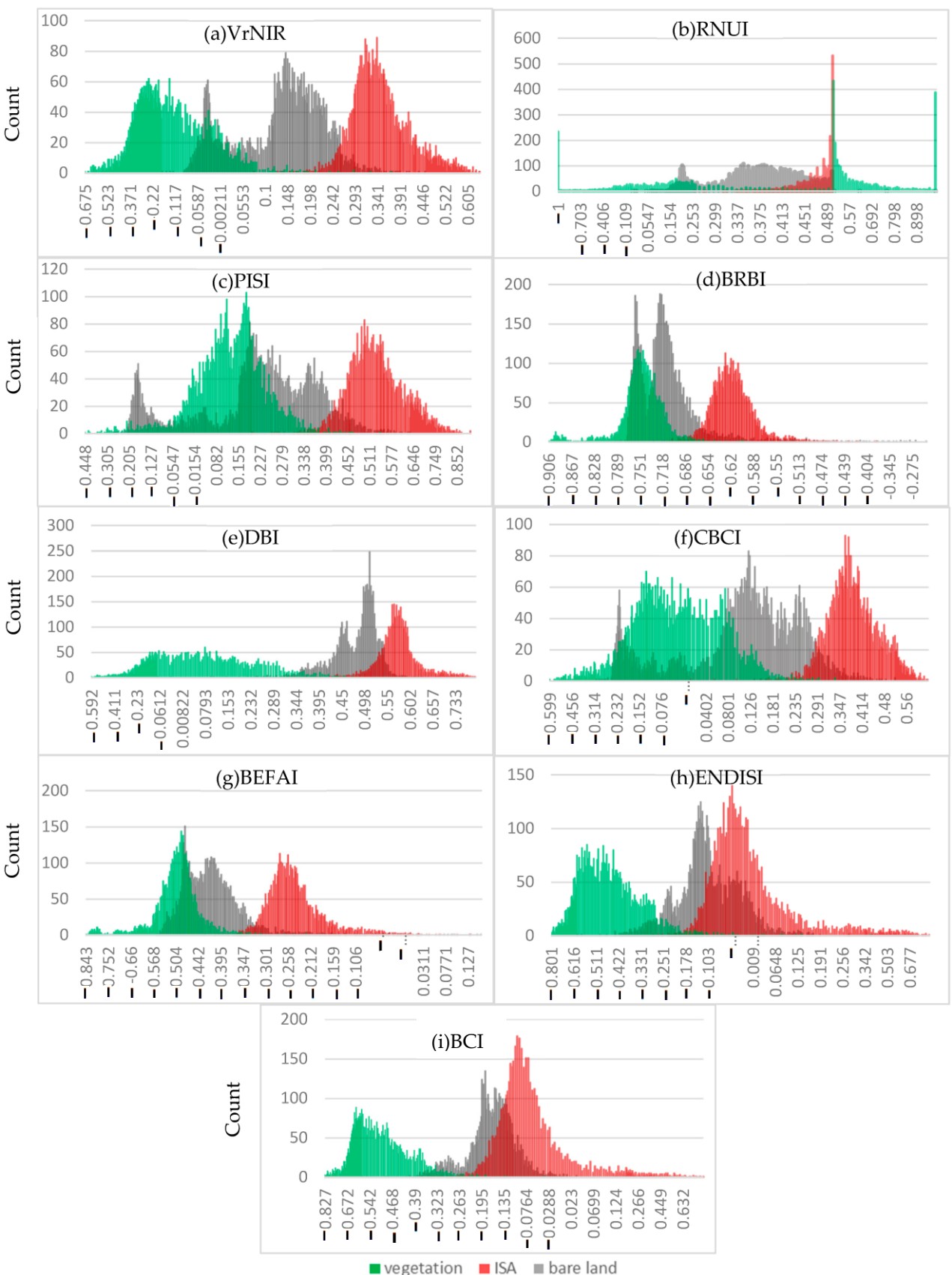

**Figure 3. Figure 3**. Histograms of land features (ISA, bare land, and vegetation) in spring for nine ISA indices in Buraydah.

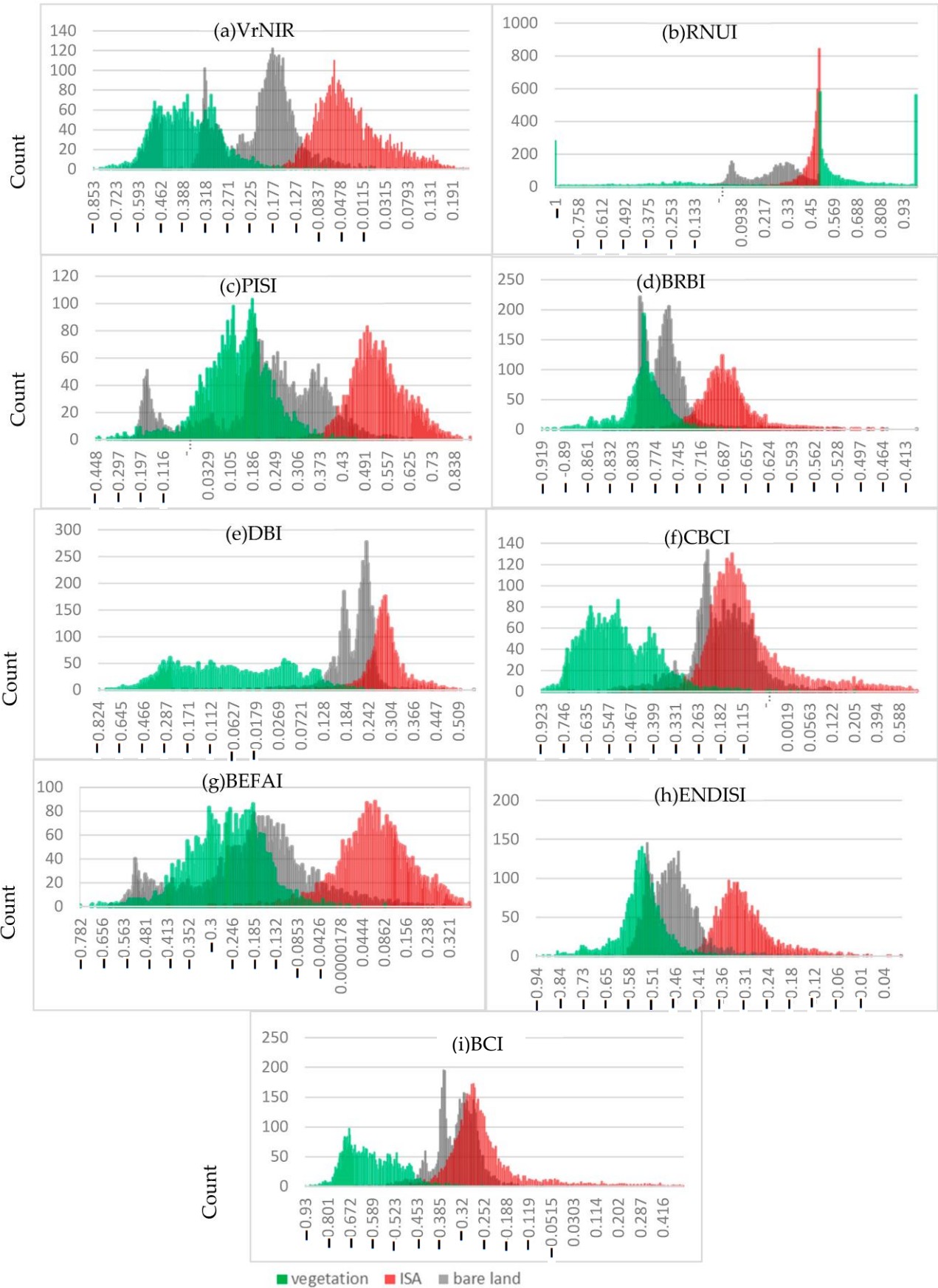

**Figure 4.** Histograms of land features (ISA, bare land, and vegetation) in summer for nine ISA indices in Buraydah.

All indicators have SDI values (1.7–2.64), J–M distance (1.155–1.387), and TD (1726–1999) in spring. In summer, chlorophyll content decreases, and plant surfaces are covered with a dust layer due to dust storms at the end of spring and the beginning of summer. All indicators have SDI (1.34–2.31), J–M distance (1.186–1.364) and TD (1453–1989) values in the summer. Notably, the VrNIR-BI index perfectly separates impervious surfaces from vegetation, while the ENDISI index is the least separative indicator between impervious surfaces and vegetation cover but generally remains good (Figures 3 and 4 and Table 3). In particular, PISI and CBCI index is the best among the studied indicators in the study area, moderately separating impervious surfaces from barren land but confusing a small amount of barren land with built-up regions. The separate results of the two indicators are very close in the two seasons, with a relative advantage for the PISI. As shown in Figures 3 and 4, the BAEI and VrNIR-BI are still a good and accurate index but also overestimate impervious areas in arid regions. The BRBI and NRUI methods are relatively less efficient than the previous indicators and have merged impervious surfaces with barren land. The ENDISI has low J–M Distance (0.642, 0.508), TD (1399), and SDI (0.59, 0.37) values in spring and summer, respectively, indicating the confusion between bare land and impervious surfaces and overestimating urbanized areas in arid lands, as shown in Figures 3 and 4.

### 3.2. ISA Classification Using Thresholding

ISA were extracted for the study area from Landsat-8 imagery acquired in spring and summer by applying the DBI, PISI, ENDISI, CBCI, BRBI, RNUI, VrNIR-BI, NRUI, and CBI indices. Since threshold segmentation requires the actual category of each pixel in the Landsat-8 images to be classified as ISA or non-ISA, it depends on how high the ISA fraction is to classify the pixel into that category, i.e., the threshold for the ISA fraction. For this reason, after calculating the index images, thresholding and unsupervised classification methods were applied to these images to distinguish ISA from non-ISA. The manual threshold was selected using a trial-and-error procedure, while the ISODATA method extracted the ISA based on classification. To determine the threshold for impervious surfaces using the ISODATA method, the following steps were performed: First, the index images were classified into ten clusters. In this algorithm, the parameters for the minimum and maximum numbers were set to 7 and 12, respectively, and default values were used for the other parameters. After the classification, derived clusters were combined into pervious and impervious classes with the help of high-resolution Google Earth images.

The values were modified a little for ranges from spring to summer due to the difference in soil moisture and the degree of the greening of plant leaves between the two seasons.

In general, the thresholding values are greater in the spring than in the summer for all indices. The derived thresholds are presented in Table 4. As shown in Table 4, the optimal thresholds selected by the ISODATA method for the index images were quite close to the values selected manually.

The results of the ISA indices were compared with the pseudo-color images (Figure 5). The ISA extraction results using threshold values are shown in Figures 6–9. Generally, these nine indices were able to detect ISA features, especially for large and homogeneous areas of ISA. Yet, the classification ability of these ISA indices is dissimilar. Most of the indicators classified the ISA more than the real ones, especially the ENDISI, BCI, and DBI, i.e., it is clearly depicted in the overview of bare soil layers. For instance, once many bare soil areas were detected using the BCI index, ISA was misclassified as bare soil. Many of the water and saline depressions have been classified as urban areas. When comparing the manual and ISOcluster thresholds, it turns out that the manual threshold is slightly better for most indices, especially in determining the roads and buildings on the outskirts of the city.

**Table 4.** ISA Classification Using Thresholding.

| ISA Indices | Summer | | Spring | |
|:---:|:---:|:---:|:---:|:---:|
| | **Manual** | **ISO Cluster** | **Manual** | **ISO Cluster** |
| CBCI | −0.0787 | −0.1354 | 0.2014 | 0.2307 |
| IPIS | 0.2014 | 0.2274 | 0.3796 | 0.4135 |
| BCI | −0.3022 | −0.3186 | −0.1192 | −0.1287 |
| VrNIR-BI | −0.1419 | −0.1101 | 0.2419 | 0.2143 |
| BLFAI | −0.4172 | −0.4321 | −0.3832 | −0.3916 |
| BRBA | −0.7333 | −0.7421 | −0.7034 | −0.6934 |
| DBI | 0.2371 | 0.2362 | 0.5216 | 0.5140 |
| ENDISI | −0.1419 | −0.1220 | −0.1216 | −0.0042 |
| NRUI | 0.4427 | 0.4321 | 0.4201 | 0.4342 |

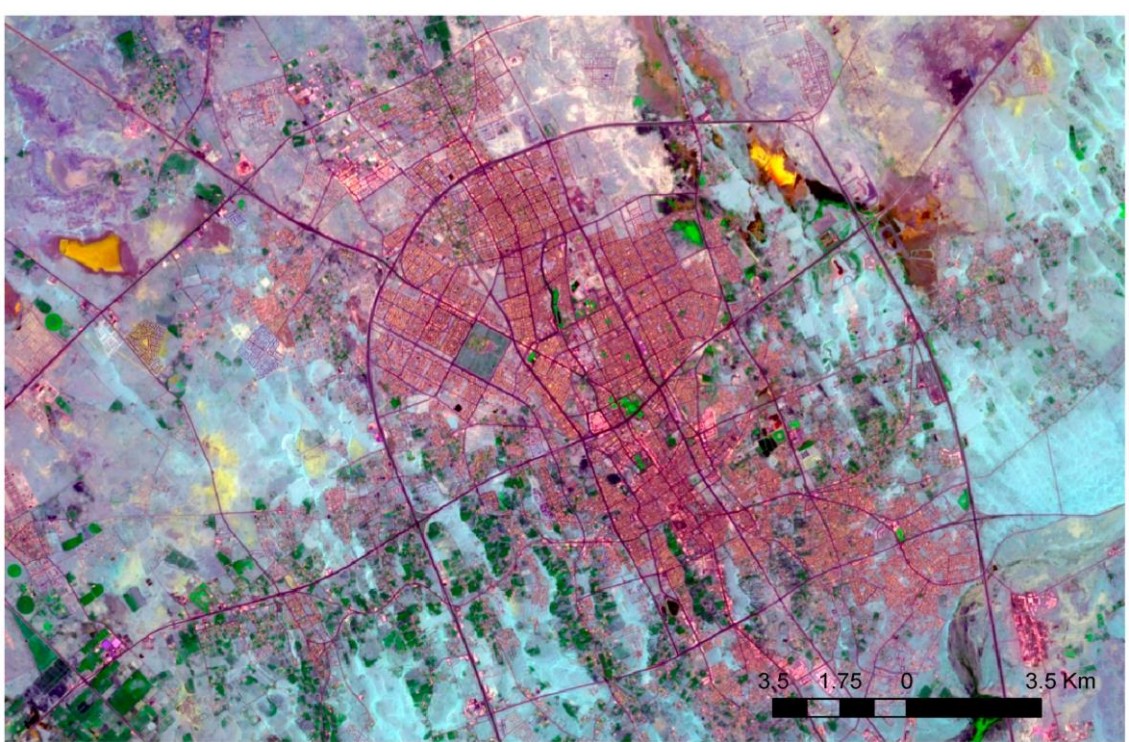

**Figure 5.** Pseudo-colors composite from Landsat-8: band 4, red; band 3, green; and band 2, blue.

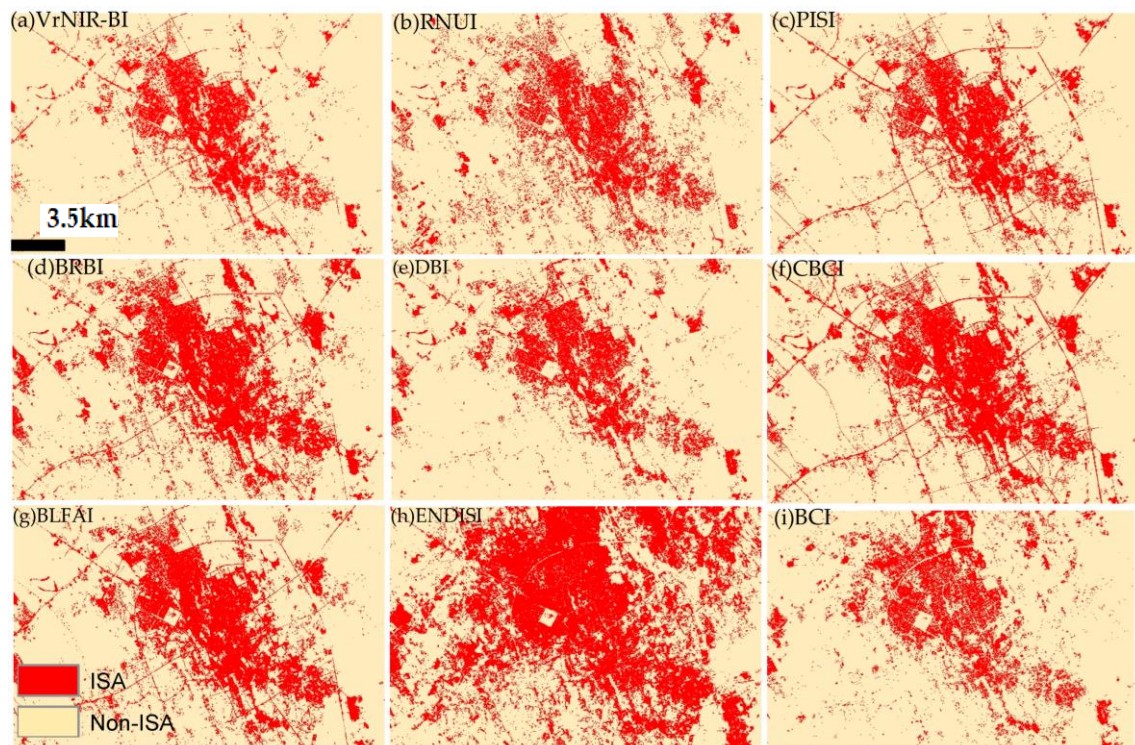

**Figure 6.** Impervious surface in spring extracted using manual thresholding for indices (**a**) VrNIR-BI, (**b**) RNUI, (**c**) PISI, (**d**) BRBI, I DBI, (**f**) CBCI, (**g**) BLFAI, (**h**) ENDISI, and (**i**) BCI.

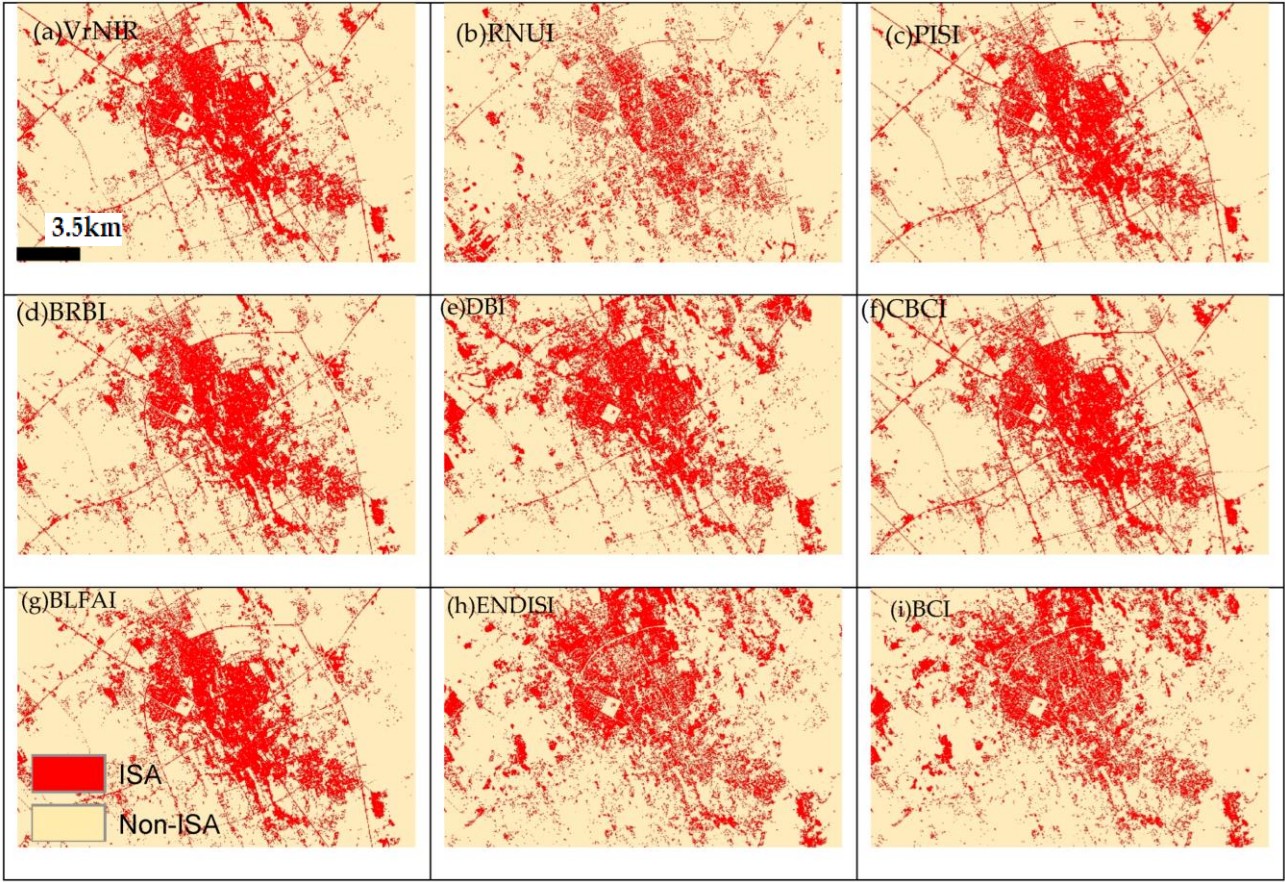

**Figure 7.** Impervious surface in summer extracted using manual thresholding for indices (**a**) VrNIR-BI, (**b**) RNUI, (**c**) PISI, (**d**) BRBI, (**e**) DBI, (**f**) CBCI, (**g**) BLFAI, (**h**) ENDISI, and (**i**) BCI.

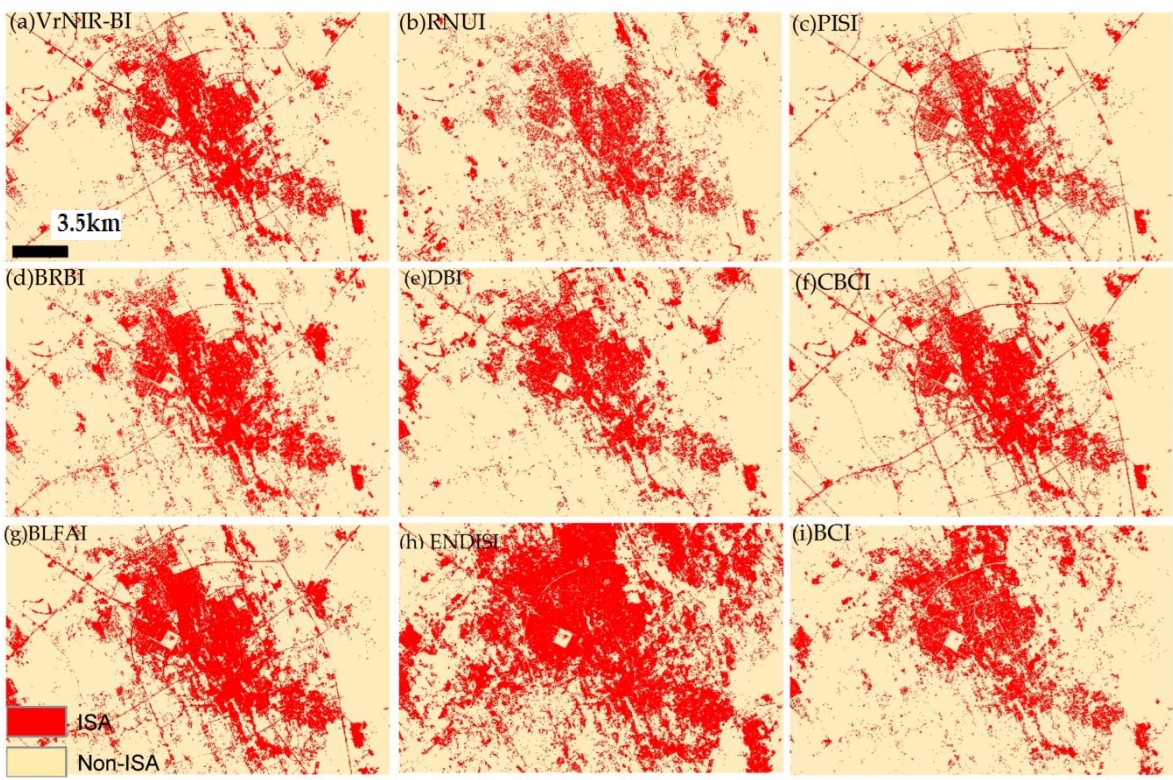

**Figure 8.** Impervious surface in spring extracted using ISODATA thresholding for indices (**a**) VgNIR, (**b**) RNUI, (**c**) PISI, (**d**) II, (**e**) DBI, (**f**) CBCI, (**g**) BLFAI, (**h**) ENDISI, and (**i**) BCI.

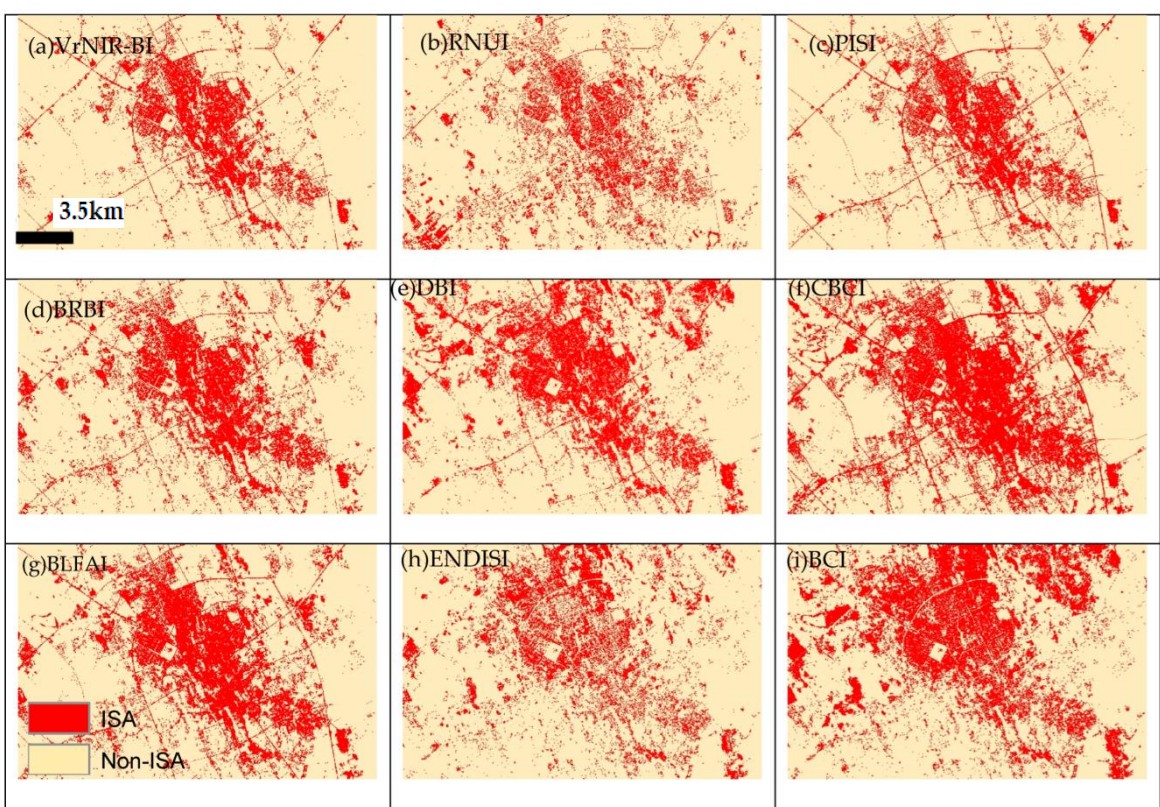

**Figure 9.** Impervious surface in summer extracted using ISO cluster thresholding for indices (**a**) VrNIR-BI, (**b**) RNUI, (**c**) PISI, (**d**) BRBI, (**e**) DBI, (**f**) CBCI, (**g**) BLFAI, (**h**) ENDISI, and (**i**) BCI.

Given the geographic location, alluvial and aeolian sediments, residential areas, and industrial structure of Buraydah city, six typical categories of impervious and pervious surfaces were selected for analysis to better understand how the categories of impervious and pervious surfaces and time affect ISA classification across different indices. These six categories of impervious and pervious surfaces include residential areas, urban–rural intersections, industrial areas, road infrastructure, dry wadi, and sand dunes.

The manual method was used to compare the results of the indicators because it is the best method for distinguishing impervious and pervious surfaces in the study area. The classification results were compared with high-resolution Google Earth images.

Valleys and depressions on the outskirts of the city of Buraydah [see Figures 10 and 11A] were formed by natural rivers and alluvium. The seasonal variation of the valleys affected the ISA extraction results. This is due to the spectral signature being confusing, with light pixels showing a similarity to industrial areas and dark pixels showing a strong similarity to shadows and road infrastructure. All ISA indices overestimated ISA because they were unable to accurately distinguish impervious surfaces from bare soil (fluvial and alluvial sediments), which is clearly seen in the ISA summary layers. In general, ENDISI and BCI performed best in the spring, while PISI and VrNIR-BI performed best in the summer.

Industrial areas [see Figures 10 and 11B] are located in the north of Buraydah. Compared to residential areas, buildings in industrial areas are less dense and show little seasonal variation. We found that VgNIR, PISI, BRBI, BLFAI, CBC, and DBI were good at extracting ISA from most images in the spring and summer. The NRUI index was able to decipher the tails of industrial buildings but could not classify roads between industrial areas as impervious surfaces. ENDISI and BCI overestimated ISA because they were unable to distinguish impervious surfaces from bare ground, especially in the spring.

Residential areas [see Figures 10 and 11C], which include a variety of land cover types, such as tall buildings, roads, vegetation, bare land, shade, etc., are located to the west of the city. The ISA layers extracted with VgNIR, PISI, BCI, RNUI, and CBC were relatively similar in spatial pattern and distribution to the visible ISA in Google Earth imagery, except for some small and scattered bare areas and some built-up areas that were misidentified.

The ENDISI shows more ISA areas than are present in reality. For example, many built-up areas and roads were identified using the ENDISI index, while some areas of bare ground were incorrectly classified as built-up.

Suburban-rural transitions [see Figures 12 and 13D] are located in the southwestern part of the city, a representative urban expansion area with significant seasonal variation in vegetation cover. Compared to the other indices, CBCI, BRBI, PISI, and BLFAI were found to be more stable to seasonal changes in vegetation and effectively separate impervious surfaces from bare ground. VrNIR-BI, DBI, and BCI underestimated ISA because they were unable to distinguish impervious surfaces from vegetation, and indicator values were closer to reality in the summer. The RNUI indicates more ISA areas than are present in reality, especially in the summer when some green areas appear as ISA areas.

Figures 12 and 13E show the extraction results for road infrastructure (e.g., highways), a precisely classified category of impervious urban surfaces. There are no significant seasonal differences between the indicators in the extraction of roads in the spring and summer. For some indicators, roads are close to reality, while for others, they are not in spring and summer. Compared to the other eight indices, the CBCI was found to be more stable to seasonal variations, outperforming the other indices and closest to ground truth. The results show that VrNIR-BI, DBI, and BCI are not able to distinguish ISA from other land cover types, while RNUI is slightly better than the other indices. The ISA layers extracted with PISI, BRBI, BAFAI, and ENDISI were relatively similar to the high-resolution Google Earth images and the visible roads in the composite images in terms of spatial patterns and distribution, except for some small and scattered road areas that were misidentified.

Figures 12 and 13F show the sand dunes on the eastern edge of Buraidah City. It is clear that all indicators are able to detect the ISA of sand dunes, which is due to the different roughness of the ISA of sand dunes. Results indicate that CBCI and PISI are more

accurate and have better separability for ISA and sand dunes than the other indices under different conditions.

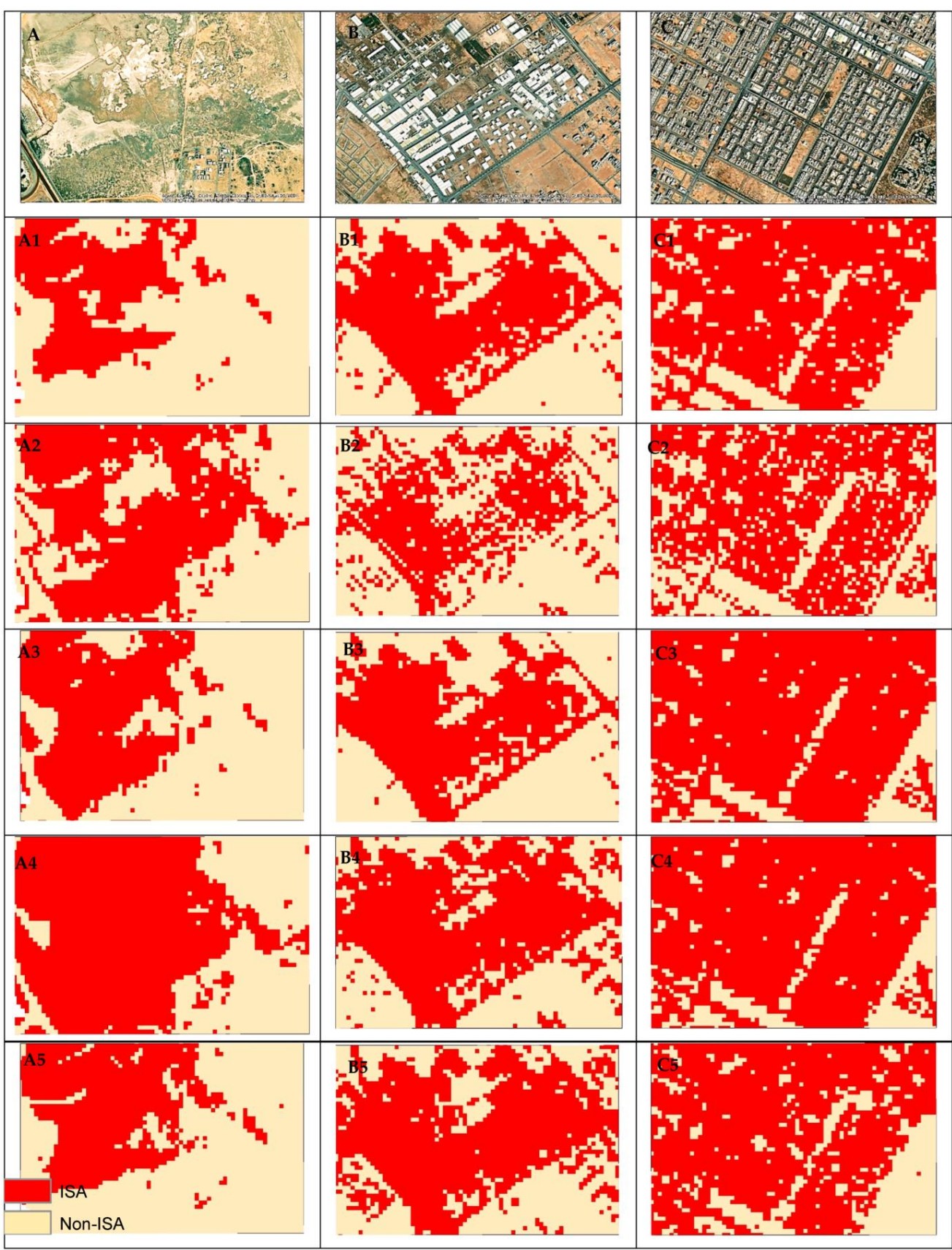

**Figure 10.** *Cont.*

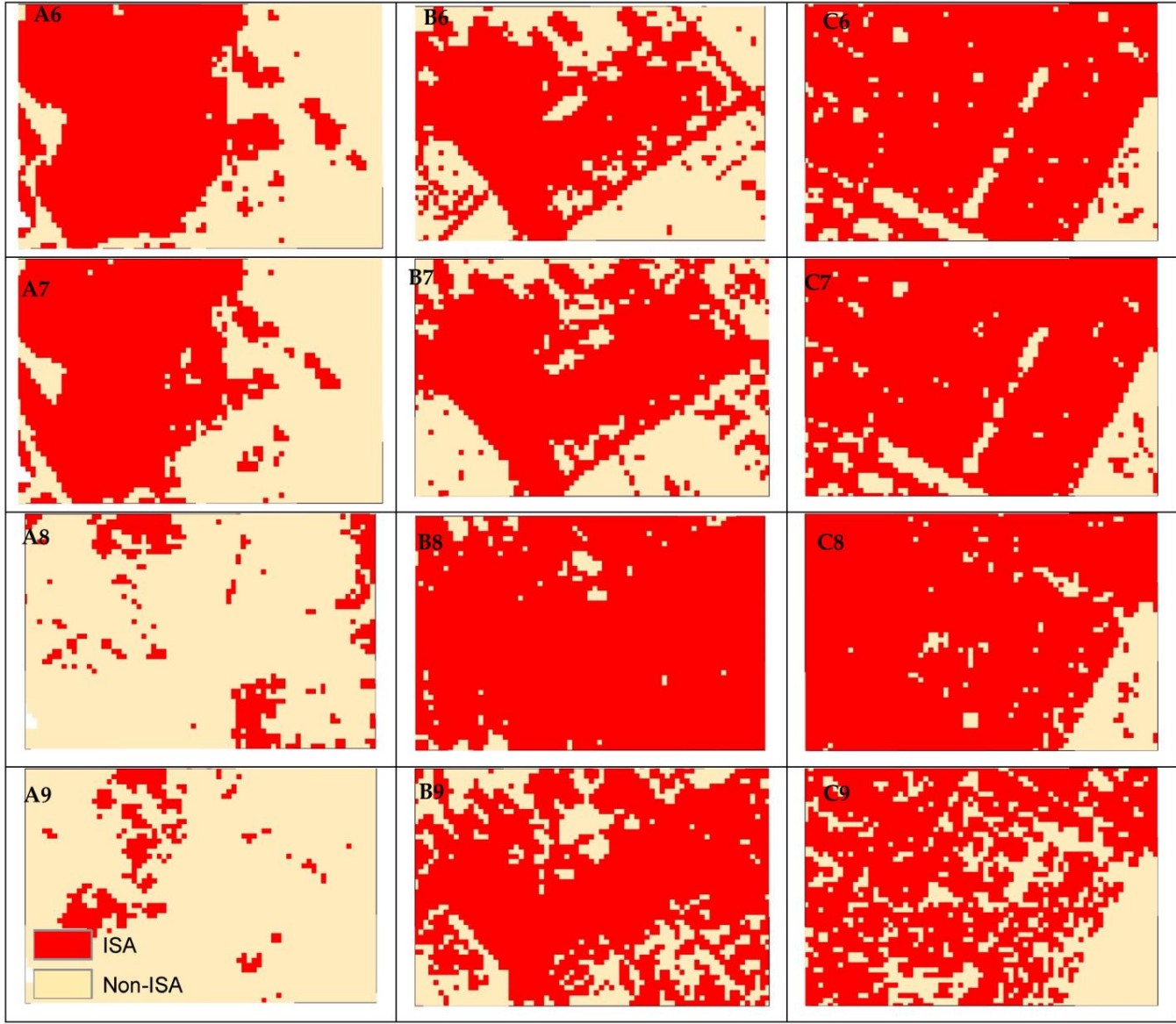

**Figure 10.** ISA distribution extracted VrNIR-BI (**A1**,**B1**,**C1**), RNUI (**A2**,**B2**,**C2**), PISI (**A3**,**B3**,**C3**), BRBI (**A4**,**B4**,**C4**), DBI (**A5**,**B5**,**C5**), CBCI (**A6**,**B6**,**C6**), BLFAI (**A7**,**B7**,**C7**), ENDISI (**A8**,**B8**,**C8**), and BCI (**A9**,**B9**,**C9**), index-based methods from spring Landsat-8 images and fine spatial resolution satellite imagery viewable in Google Earth Pro in the three subsets of the study area: Valleys and depressions on the outskirts of the city (**A**); industrial area (**B**); residential area (**C**).

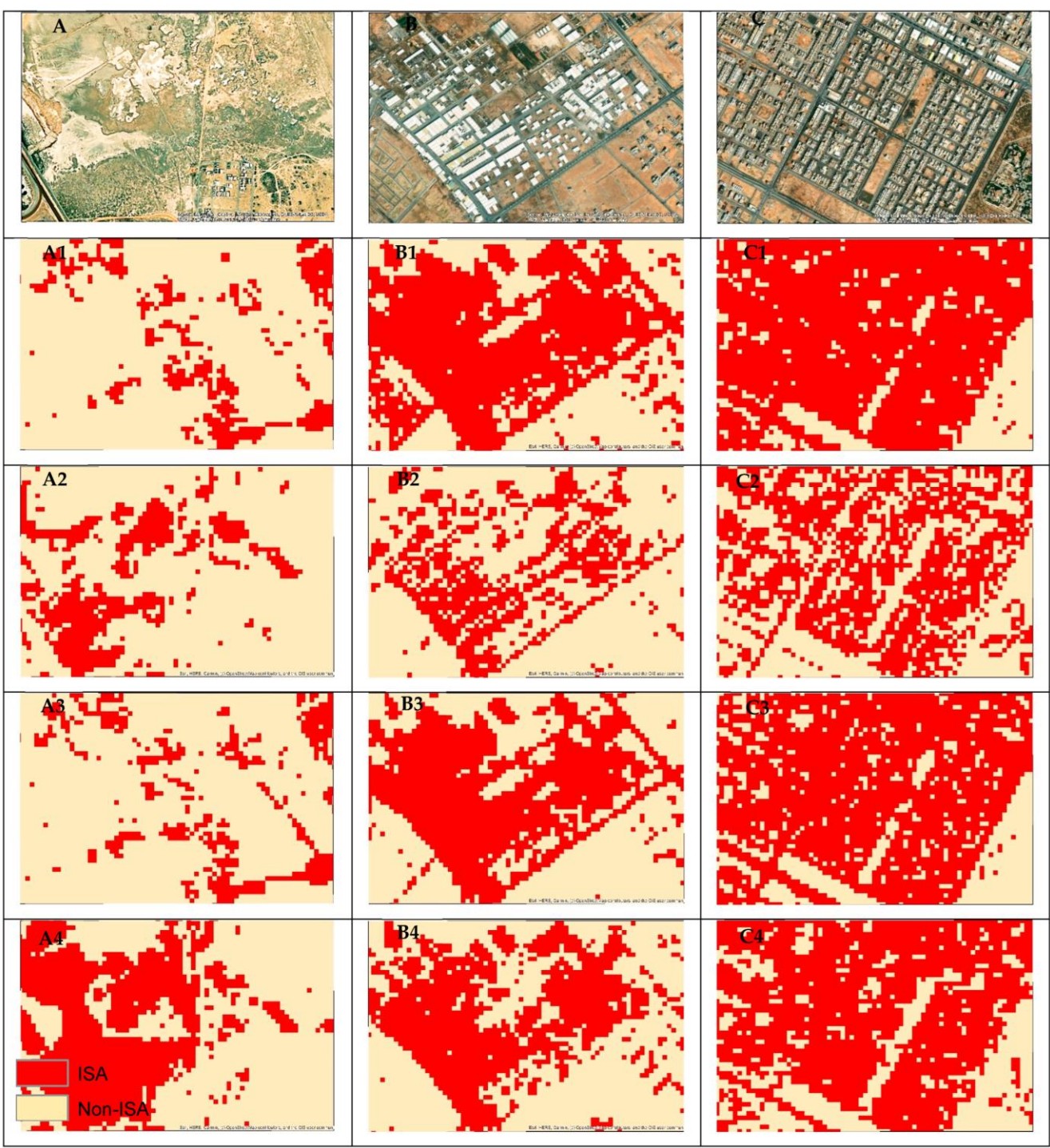

**Figure 11.** *Cont.*

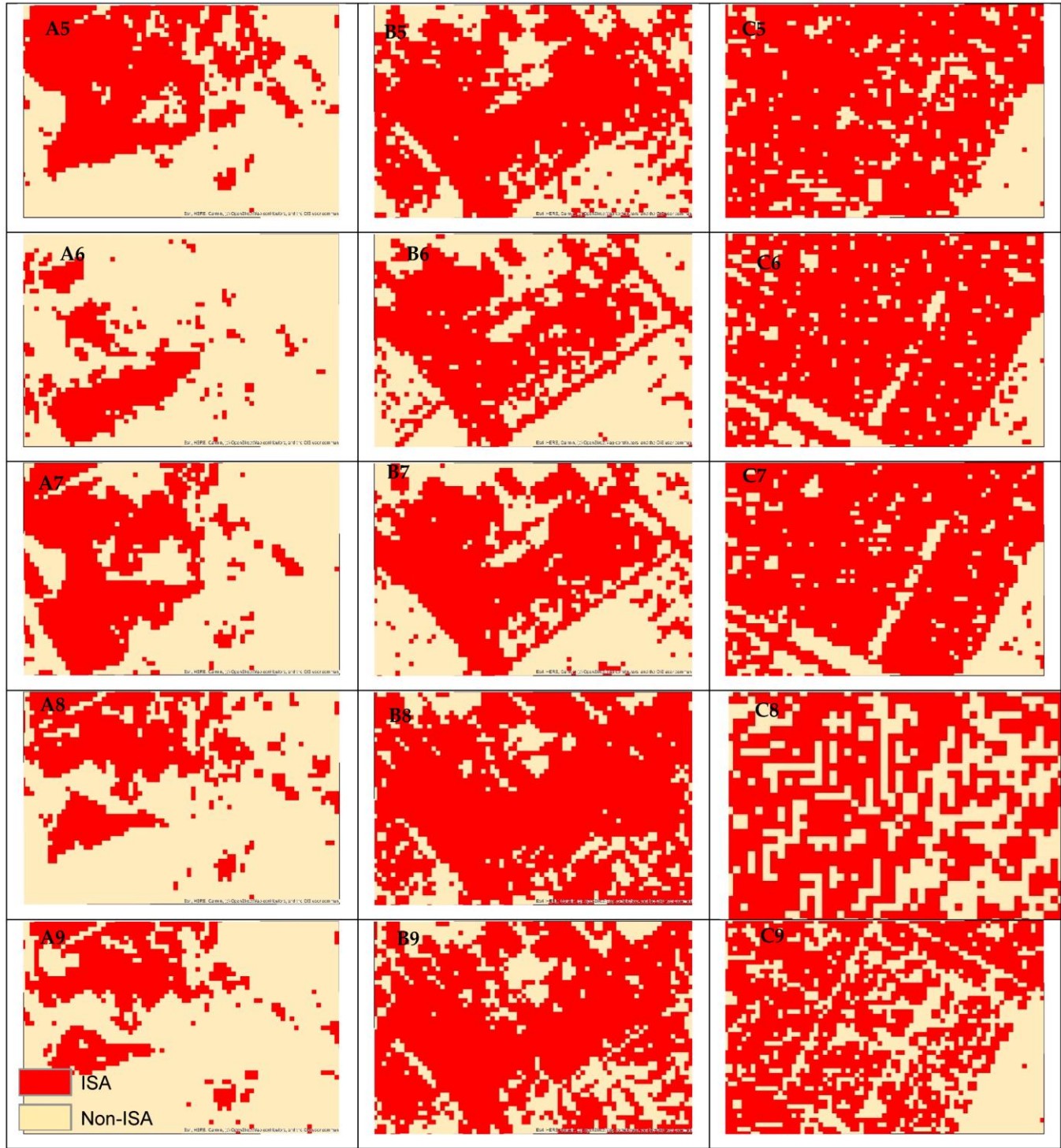

**Figure 11.** ISA distribution extracted VrNIR-BI (**A1,B1,C1**), RNUI (**A2,B2,C2**), PISI (**A3,B3,C3**), BRBI (**A4,B4,C4**), DBI (**A5,B5,C5**), CBCI (**A6,B6,C6**), BLFAI (**A7,B7,C7**), ENDISI (**A8,B8,C8**) and BCI (**A9,B9,C9**) index-based methods from summer Landsat-8 images and fine spatial resolution satellite imagery viewable in Google Earth Pro in the three subsets of the study area: Valleys and depressions on the outskirts of the city (**A**); industrial area (**B**); residential area (**C**).

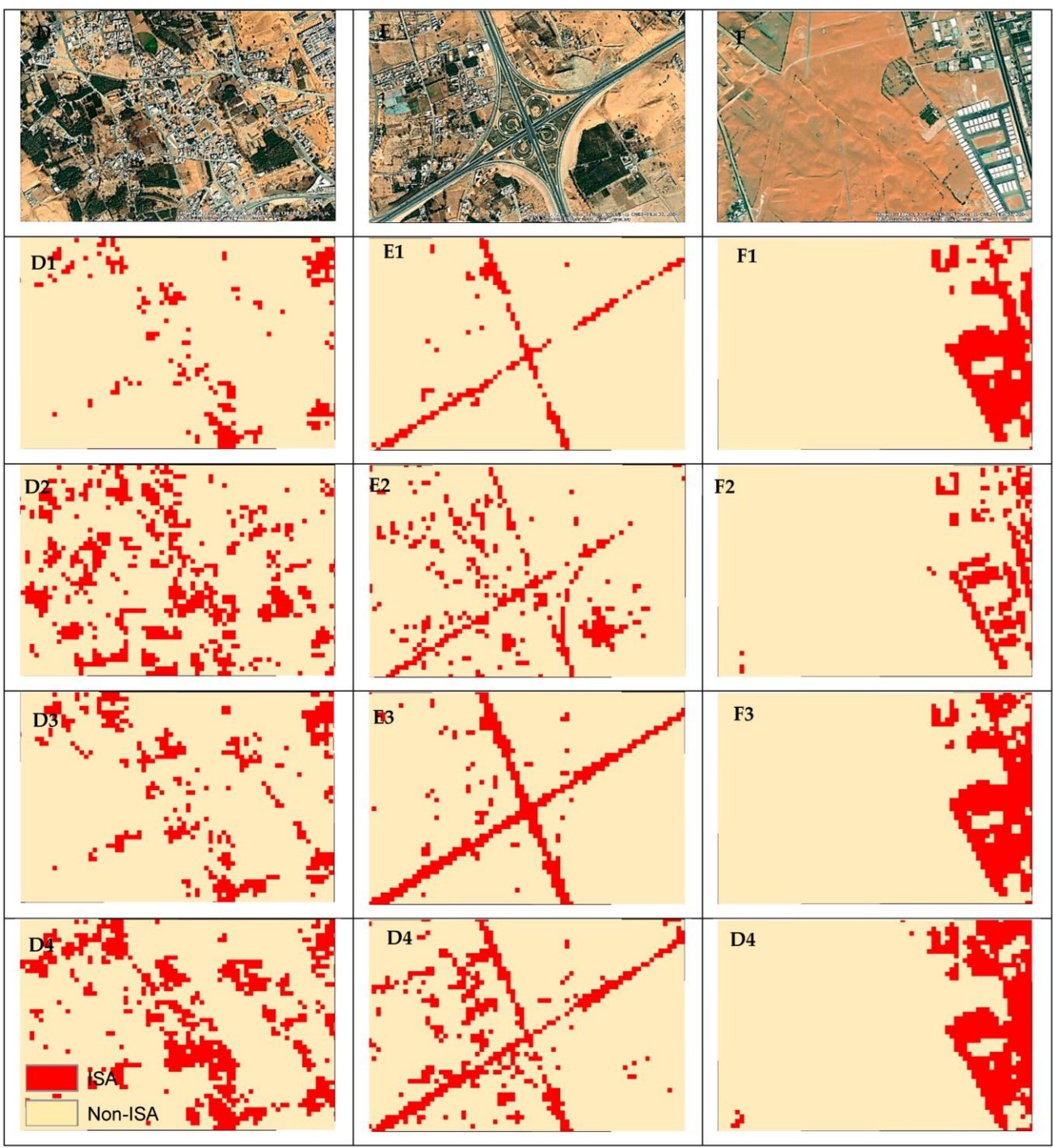

**Figure 12.** *Cont.*

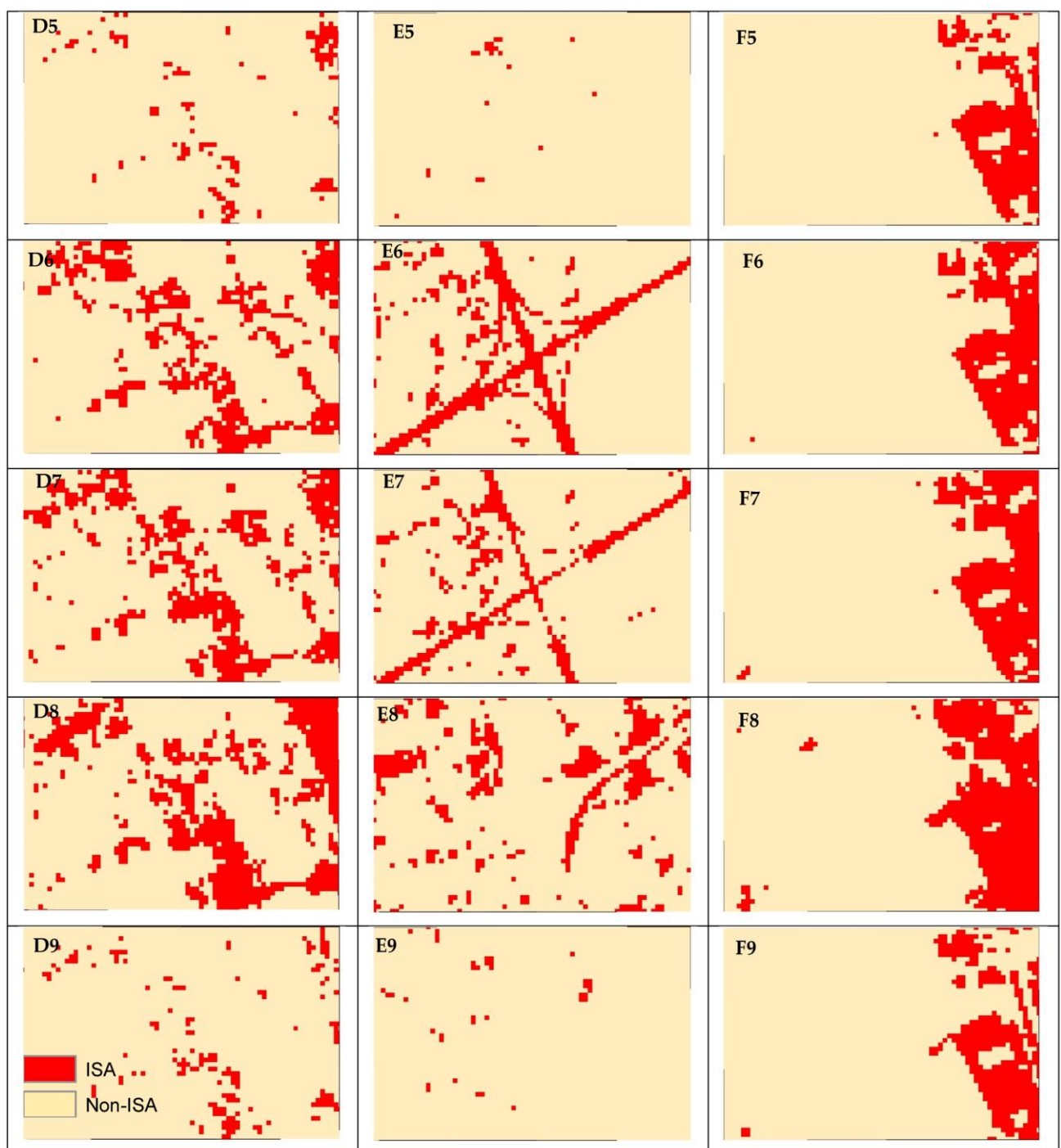

**Figure 12.** ISA distribution extracted VrNIR-BI (**D1**,**E1**,**F1**), RNUI (**D2**,**E2**,**F2**), PISI (**D3**,**E3**,**F3**), BRBI (**D4**,**E4**,**F4**), DBI (**D5**,**E5**,**F5**), CBCI (**D6**,**E6**,**F6**), BLFAI (**D7**,**E7**,**F7**), ENDISI (**D8**,**E8**,**F8**), and BCI (**D9**,**E9**,**F9**), index-based methods from spring Landsat-8 images and fine spatial resolution satellite imagery viewable in Google Earth Pro in the three subsets of the study area: suburban-rural transitions (**D**); highways (**E**); and sand dunes (**F**).

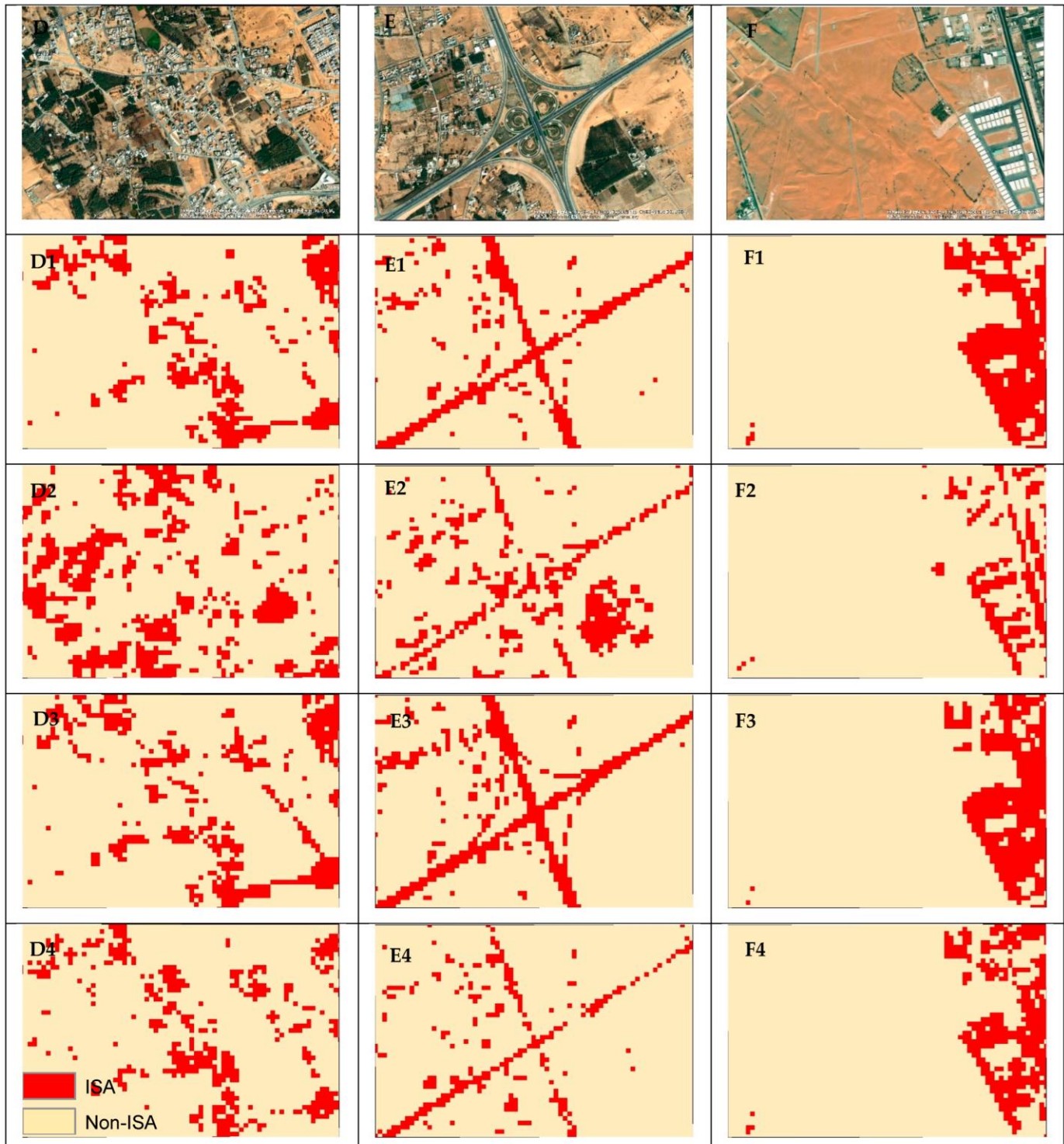

**Figure 13.** *Cont.*

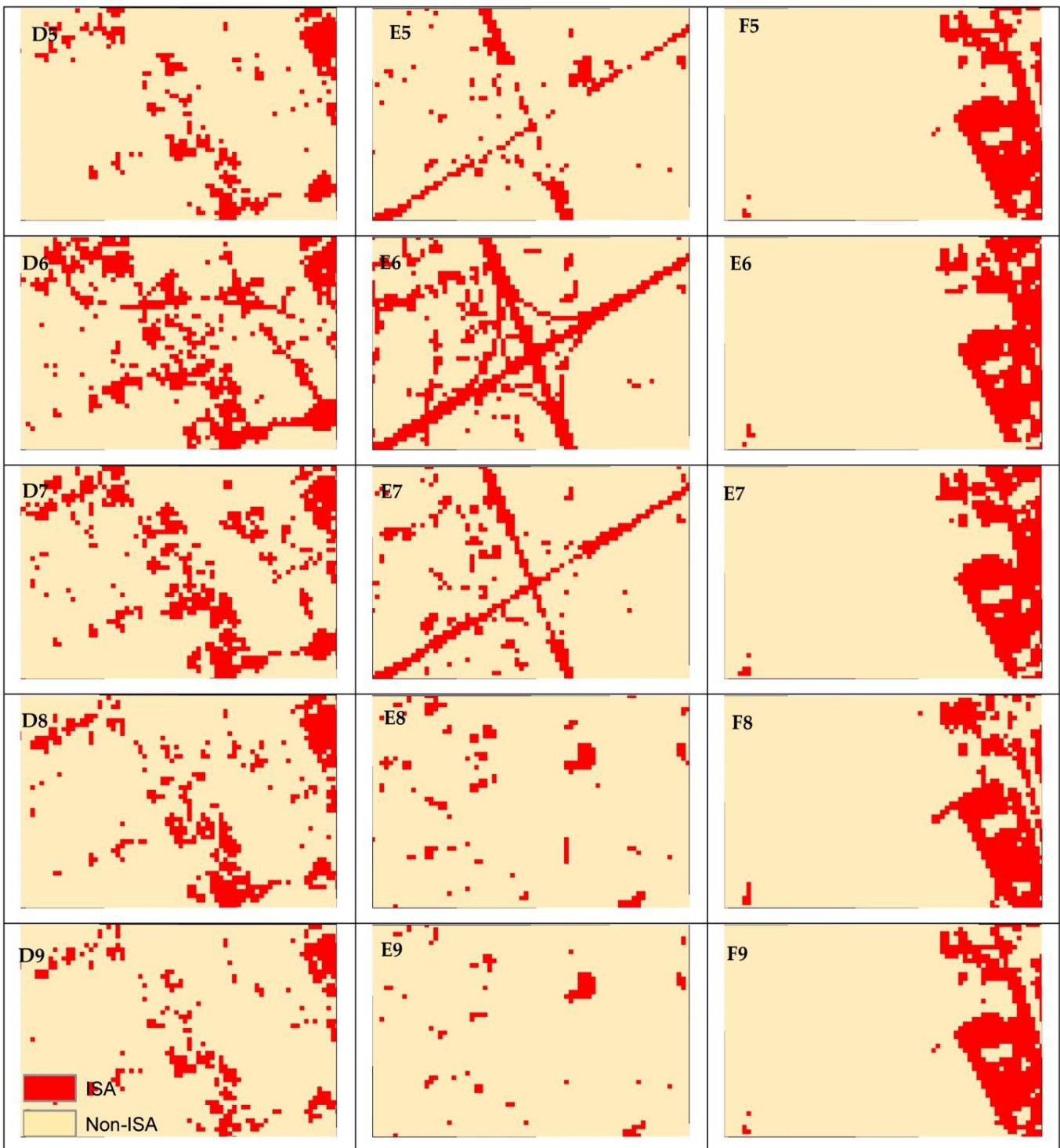

**Figure 13.** ISA distribution extracted VrNIR-BI (**D1,E1,F1**), RNUI (**D2,E2,F2**), PISI (**D3,E3,F3**), BRBI (**D4,E4,F4**), DBI (**D5,E5,F5**), CBCI (**D6,E6,F6**), BLFAI (**D7,E7,F7**), ENDISI (**D8,E8,F8**), and BCI (**D9,E9,F9**), index-based methods from summer Landsat-8 images and fine spatial resolution satellite imagery viewable in Google Earth Pro in the three subsets of the study area: suburban-rural transitions (**D**); highways (**E**); sand dunes (**F**).

### 3.3. Performance Assessment

To quantitatively evaluate the accuracy of the nine ISA indices for two seasonal images, we selected a total of 300 samples for the impervious (150 samples) and previous (150 samples) classes. The ISA layers were compared to ground-truth points (i.e., the second

dataset of pure pixels). The results were evaluated mainly using the confusion matrix, overall accuracy (OA), and kappa as evaluation indicators. Table 5 shows the evaluation of accuracy. The results of the study show that the manual method and the ISODATA method provide similar results, with the manual method being relatively preferred. Therefore, we will use the manual method to compare the performance of the nine tested indices in the spring and summer. The PISI and CBCI indices performed best in classifying ISA compared to the other indices, with an average OA of 88.5% and 88% and a kappa of 0.76 and 0.75, respectively. However, the PISI index is better than the CBCI index in summer, with an OA of 88.5% and a kappa of 0.75, and the CBCI index is better in spring, with an OA of 89.5% and a kappa of 0.79 (Table 5). The difference in assessment scores between summer and spring is also smaller for the PISI. The VrNIR-BI index had the second-highest performance in detecting ISA, with an average OA of 85.7% and a kappa of 0.73.

**Table 5.** Accuracy assessment of ISA features extracted by nine ISA indices.

| ISA Indices | Accuracy | Manual | | | ISO Data | | |
|---|---|---|---|---|---|---|---|
| | | Spring | Summer | Average | Spring | Summer | Average |
| VrNIR-BI | OA (%) | 88.5 | 86.5 | 87.5 | 86 | 85.5 | 85.75 |
| | Kappa | 0.77 | 0.75 | 0.76 | 0.72 | 0.73 | 0.725 |
| RNUI | OA (%) | 79 | 78 | 78.5 | 78 | 79.5 | 78.75 |
| | Kappa | 0.6 | 0.59 | 0.595 | 0.56 | 0.59 | 0.575 |
| PISI | OA (%) | 89 | 88 | 88.5 | 87.5 | 85.5 | 86.5 |
| | Kappa | 0.77 | 0.75 | 0.76 | 0.75 | 0.71 | 0.73 |
| BRBI | OA (%) | 83 | 85 | 84.5 | 84.5 | 83.5 | 84 |
| | Kappa | 0.69 | 0.71 | 0.7 | 0.7 | 0.69 | 0.695 |
| DBI | OA (%) | 76.5 | 74.5 | 76 | 74.5 | 73.5 | 74 |
| | Kappa | 0.53 | 0.52 | 0.525 | 0.52 | 0.51 | 0.515 |
| CBCI | OA (%) | 88.5 | 87.5 | 88 | 87 | 86.5 | 86.75 |
| | Kappa | 0.77 | 0.73 | 0.76 | 0.74 | 0.73 | 0.735 |
| BLFAI | OA (%) | 85 | 82.5 | 83.75 | 84.5 | 83 | 83.75 |
| | Kappa | 0.7 | 0.65 | 0.675 | 0.69 | 0.66 | 0.675 |
| ENDISI | OA (%) | 76 | 75 | 75.5 | 69.5 | 72.5 | 71 |
| | Kappa | 0.52 | 0.5 | 0.51 | 0.39 | 0.45 | 0.42 |
| BCI | OA (%) | 78 | 77 | 77.5 | 77 | 76.5 | 76.75 |
| | Kappa | 0.56 | 0.55 | 0.555 | 0.57 | 0.57 | 0.57 |

The ENDISI, DBI, and BCI indices did not perform well compared to the other indices in classifying ISA, with an average OA of 75.5%, 76%, and 77.5% and a kappa of 0.51, 0.53, and 0.56, respectively. These indicators were not able to separate the sandy soils from the buildings in the east of the study area, while the results of the indices BRBI, BLFAI, and RNUI (with an average OA of 84.5%, 83.75%, 78% and a kappa of 0.7, 0.68, and 0.60, respectively) were the most effective compared to the indices ENDISI, DBI, and BCI (Table 5).

## 4. Discussion

In this paper, we compared and evaluated the performance of VrNIR-BI, RNUI, PISI, BRBI, DBI, CBCI, BLFAI, ENDISI, and BCI in extracting ISA from Landsat-8 imagery in a dry area (Buraydah), as well as the effects of relatively wet periods (spring) and dry periods (summer) on the ISA extraction results. This apart, two impervious surface binary methods (the manual thresholding method and the ISODATA classification method) were tested

on multiseasonal Landsat-8 images in the main urban area of Buraydah, Saudi Arabia. Although ISA index-based methods are quick and easy to implement, their efficiency varies depending on the characteristics of the climate patterns, and The ISA index is a relatively sensitive indicator. The results confirmed that when using Landsat 8 OLI-TIRS data for ISA extraction, some ISA indices performed better than others.

The results of the manual method and the ISODATA classification method are similar, with the manual method having a relative advantage. A visual inspection of the images indicates in the histogram that the confusion of bare soil and ISA has increased significantly with respect to the images obtained in summer, where low plant density is presented. In fact, this result confirms that built-up indices provide better results for seasons where rain or humidity is present in the study area, and this is consistent with what was stated in previous studies, such as Stathakis et al. [50], Sun et al. [32], Valdiviezo-N et al. [51].

Rasul [4] also pointed out that the index of impervious surface performs differently in humid and dry-arid regions. For example, the DBI index was found to be very accurate in Erbil (Iraq), but it is an inappropriate indicator for mapping bare surfaces in our test area. Indeed, the region where the DBI index was applied belongs to the semi-arid climate in Erbil and not to the arid climate, as is the case in our study area. Moreover, the DBI index depends on the thermal channel, and the results of this index depend on the acquisition time of the satellite image during the day. If the image was taken three hours after sunrise, as is the case in our study area, the thermal differences between the urban and the sandy arid areas are small, leading to confusion between the urban areas and the sandy areas.

The RNUI method has largely merged urbanized zones with barren land; however, it has largely separated the vegetative area from the barren class. However, it was found through this study that this item is not suitable for separating built and dry land in a dry climate. Chen et al. [41] and Chen et al. [6] indicated that the ENDISI effectively distinguished impervious surface from the background in subtropical monsoon climate areas, but it is an inappropriate indicator for ISA mapping in our study area (i.e., dry climate region). However, ENDISI values overlapped with bare soil and grass on dark impervious surfaces. We should consider the study area's climatic conditions, as well as its effectiveness, when choosing an appropriate ISA index. In addition, surface features should generally be considered when using spectral indices. The value range of ISA area and bareness in ENDISI has larger overlapping zones in comparison to other indices in our study area.

PISI outperforms the other eight indices, especially with the extensive existence of ISA. Such findings are not surprising, as they are consistent with the results reported by Tian et al. [34] and Li et al. [37]. Experimental results show that PISI and CBCI are relatively robust to seasonality when extracting ISA from different images. The strength of CBCI is that it can distinguish ISA when mixed with vegetation areas but is not quite good with wet soil areas, fluvial sediments, and aeolian sediments.

The VrNIR-BI, BAEI and BRBI indices are still relatively good, accurate indices, but they also overestimate ISA, and barren land is not visible between the buildings in some places, as shown in Figures 6 and 7. In this study, it is found that, unlike the study of Li et al. [37], there is no clear effect of shadow on the accuracy of the results, which is due to the average discrimination ability of Landsat-8 images in addition to the low buildings in the city, which are between 1–3 floors.

Ultimately, some indicators provided good results, such as the PISI and CBCI indicators for determining ISA. Nevertheless, future studies should continue to search for an index or technique to complete the distinction between ISA and wet bare soil, fluvial sediments, and aeolian sediment for research in urban, semi-urban, and rural areas in dry climate areas.

### 5. Conclusions

Extracting ISA from satellite images with medium spectral and spatial resolution, like Landsat 8 is not an easy task. Accurate extraction of ISA is critical for municipal construction, urban planning, sustainable development, and environmental assessment. The existing methods, such as mixture analysis methods and classification-based methods, the former is highly dependent on the quality of the end members, and the process for capturing the end members is very complicated for some of these methods. For the latter, they require high-quality training data. Index-based methods, on the other hand, are easy to implement and have acceptable accuracy.

The availability of free Landsat-8 data opens new opportunities for mapping large-scale impervious surfaces. This is the first study to compare the performance of different spectral data for ISA extraction using two seasonal Landsat-8 images in arid climates. Using two seasonal Landsat-8 images from the urban area of Burydah, Saudi Arabia, nine index-based methods for detecting impervious surfaces (PISI, BCI, VrNIR-BI, BAEI, BRBI, DBI, RNUI, ENDISI, and CBCI) and two binary methods for detecting impervious surfaces (manual thresholding method and ISODATA classification method) were tested. Histogram overlap method, SDI, J–M distance, TD, OA, and kappa values were used to evaluate the results. The ISA index is a relatively sensitive indicator. Although index-based methods are fast and easy to implement, their efficiency varies depending on the characteristics of the climate patterns.

Four conclusions can be drawn from this study: PISI and CBCI are best at distinguishing impervious surfaces from non-impervious surfaces in Landsat-8 imagery in dry climates compared to the other seven indices, with PISI significantly reducing misclassification between ISA and bare soil and vegetation features. PISI is a more reliable index. Second, DBI and ENDISI extracted impervious surfaces in summer and spring in dry climates worse than the other indices. Although DBI was applied in a semi-arid climate in Erbil city and proved to be effective, it is not effective in an arid climate, such as that of our study area. Third, calculating built-up indices from image sets taken in dry months made it more difficult to distinguish bare ground from urban areas for most index-based methods. This result confirms that ISA indices provide better results in months when it is relatively wet in the study area. The ISODATA classification method and manual thresholding have similar results when generating ISA maps, but the manual method is a little better than ISODATA at achieving stable ISA mapping. Although the manual method performs slightly better in the study area, the automated method (ISODATA) remains effective and accurate and can be used to quickly and automatically select a threshold that is similar to the optimal threshold.

This paper serves as a reference for selecting spectral indices for ISA extraction from Landsat-8 imagery based on climatic conditions and for selecting an appropriate binary processing method. This research contributes to the further study and application of Landsat-8 data. The study suffers from some possible limitations because the ISA indices were applied to only one city, and it is necessary to test them in other regions with dry climates and other land cover characteristics. However, the results of this study can be further used to identify the impervious urban characteristics of similar geoclimatic and urban conditions.

In future work, we should keep in mind that the ISA indices were only applied to one city and that they need to be tested in other regions with dry climates and different land cover characteristics. For example, in the cities of Mecca and Medina, whose surfaces in and around the cities consist of basalt, which is very similar to the surfaces of roads and parking lots. Another point: the authors intend to develop a new specific ISA index for dry climate.

**Author Contributions:** Conceptualization, H.A.; methodology, H.A. and I.O.A.; software, H.A.; validation, H.A. and I.O.A.; formal analysis, H.A.; investigation, H.A. and I.O.A.; resources, H.A.; data curation, H.A.; writing—original draft preparation, H.A.; writing—review and editing, H.A. and I.O.A.; visualization, H.A.; supervision, H.A.; project administration, H.A.; funding acquisition, I.O.A. All authors have read and agreed to the published version of the manuscript.

**Funding:** The article processing charge was funded by the Deanship of Scientific Research, Qassim University.

**Institutional Review Board Statement:** Not applicable.

**Informed Consent Statement:** Not applicable.

**Data Availability Statement:** Not applicable.

**Acknowledgments:** Researchers would like to thank the Deanship of Scientific Research, Qassim University, for funding the publication of this project.

**Conflicts of Interest:** The authors declare no conflict of interest.

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
