# Peer review of "Evaluation of Index-Based Methods for Impervious Surface Mapping from Landsat-8 to Cities in Dry Climates; A Case Study of Buraydah City, KSA"

_sustainability, doi:10.3390/su15129704_

Round 1

Reviewer 1 Report

The authors performed  mapping impervious surface area (ISA) for a city in dry climate. They used two unsupervised classification methods. The first manual and the second automated.

The authors should explain the reason they did not perform a supervised classification.

The meaning of the colors in fig.2 and fig. 3 is not clear.

Please correct the captions of a couple of figures that are barely understandable

Figure 10. ISA distribution extracted VrNIR (A1, B1, C1), RNUI (A2, B2, C2), PISI (A3, B3, C3), BRBI, DBI, CBCI, BLFAI, ENDISI, and 654
BCI index-based methods from summer Landsat-8 images and fine spatial resolution satellite imagery viewable in Google Earth Pro in the three subsets of the study area: Valleys and depressions on the outskirts of the city (A); industrial area (B); residential 656
area (C).

There is no explanation for panels from A4 to A9, B4 to B9, C4 to C9

Caption fig. 12 says:

ISA distribution extracted VrNIR (D1, E1, F1), RNUI (D2, B2, C2), PISI (D3, B3, C3), BRBI, DBI, CBCI, BLFAI, ENDISI, and 673
BCI index-based methods from summer Landsat-8 images and fine spatial resolution satellite imagery viewable in Google Earth Pro 674
in the three subsets of the study area: suburban-rural transitions (D); highways (E); sand dunes (F).

But panels go from A to A9, B to B9 and C to C9. Thus I, the reader, do not understand where D; E and F live.

Author Response

Dear reviewer 1, First of all, we would like to thank you for giving a timely review report/feedback.   The comments you have given are really very effective and comprehensive. We are grateful to you for helping us to improve our manuscript.  Below we provide detailed point to point responses to all comments and changes made. We hope our revisions are satisfactory and our paper can be accepted for publication.

All corrections in the manuscript are done by highlighted in “red colour text”.

1-The authors performed  mapping impervious surface area (ISA) for a city in dry climate. They used two unsupervised classification methods. The first manual and the second automated. The authors should explain the reason they did not perform a supervised classification.

Replay: The objective was to evaluate the effectiveness of automated thresholding procedures for extracting ISA in a study area. The study sought to determine whether these methods could be used to rapidly and automatically select a threshold as an effective tool for large-scale monitoring of ISA changes, while eliminating the uncertainties associated with manual threshold selection for effective extraction.

Large number of papers has used automatic methods to determine the optimal threshold such as:

Li, C.; Shao, Z.; Zhang, L.; Huang, X.; Zhang, M. A Comparative Analysis of Index-Based Methods for Impervious Surface Mapping Using Multiseasonal Sentinel-2 Satellite Data. IEEE J. Sel. Top. Appl. Earth Obs. Remote Sens. 2021, 14, 3682–3694, doi:10.1109/JSTARS.2021.3067325.

Bouhennache, R.; Bouden, T.; Taleb-Ahmed, A.; Cheddad, A. A New Spectral Index for the Extraction of Built-up Land Features from Landsat 8 Satellite Imagery. Geocarto Int. 2019, 34, 1531–1551, doi:10.1080/10106049.2018.1497094.

Li, H.; Wang, C.; Zhong, C.; Su, A.; Xiong, C.; Wang, J.; Liu, J. Mapping Urban Bare Land Automatically from Landsat Imagery with a Simple Index. Remote Sens. 2017, 9, doi:10.3390/RS9030249.

Lu, D.; Hetrick, S.; Moran, E. Impervious Surface Mapping with Quickbird Imagery. Int. J. Remote Sens. 2011, 32, 2519–2533, doi:10.1080/01431161003698393.

Li, H.; Wang, C.; Zhong, C.; Su, A.; Xiong, C.; Wang, J.; Liu, J. Mapping Urban Bare Land Automatically from Landsat Imagery with a Simple Index. Remote Sens. 2017, 9, doi:10.3390/rs9030249.

Bhatti, S.S.; Tripathi, N.K. Built-up Area Extraction Using Landsat 8 OLI Imagery. GIScience Remote Sens. 2014, 51, 445–467, doi:10.1080/15481603.2014.939539.

Sun, Z.; Wang, C.; Guo, H.; Shang, R. A Modified Normalized Difference Impervious Surface Index (MNDISI) for Automatic Urban Mapping from Landsat Imagery. Remote Sens. 2017, 9, doi:10.3390/RS9090942.

Bouhennache, R.; Bouden, T.; Taleb-Ahmed, A.; Cheddad, A. A New Spectral Index for the Extraction of Built-up Land Features from Landsat 8 Satellite Imagery. Geocarto Int. 2019, 34, 1531–1551, doi:10.1080/10106049.2018.1497094.

2-The meaning of the colors in fig.2 and fig. 3 is not clear.

Reply: The legend has been added.

3-Please correct the captions of a couple of figures that are barely understandable

 Reply: As suggested by the reviewer, the figures have been modified and corrected.

4-Figure 10. ISA distribution extracted VrNIR (A1, B1, C1), RNUI (A2, B2, C2), PISI (A3, B3, C3), BRBI, DBI, CBCI, BLFAI, ENDISI, and 654
BCI index-based methods from summer Landsat-8 images and fine spatial resolution satellite imagery viewable in Google Earth Pro in the three subsets of the study area: Valleys and depressions on the outskirts of the city (A); industrial area (B); residential 656
area (C).

There is no explanation for panels from A4 to A9, B4 to B9, C4 to C9

 Reply: As suggested by the reviewer, the figures have been modified and corrected.

5-Caption fig. 12 says:

ISA distribution extracted VrNIR (D1, E1, F1), RNUI (D2, B2, C2), PISI (D3, B3, C3), BRBI, DBI, CBCI, BLFAI, ENDISI, and 673
BCI index-based methods from summer Landsat-8 images and fine spatial resolution satellite imagery viewable in Google Earth Pro 674
in the three subsets of the study area: suburban-rural transitions (D); highways (E); sand dunes (F).

But panels go from A to A9, B to B9 and C to C9. Thus I, the reader, do not understand where D; E and F live.

 Reply: As suggested by the reviewer, The figures have been modified and corrected.

The authors are also grateful to anonymous reviewers and the editor for their constructive comments which improved the manuscript.

Reviewer 2 Report

Dear authors.

I find the article intriguing as it presents promising prospects for advancements in this area through innovative solutions that can be applied to real-world situations. The article exhibits acceptable structure and well-written content. However, there are several crucial aspects that have either been inadequately explained or are noticeably absent from the text. Certain aspects have been minimized or left unexplained. Please find my suggestions my comments in the attached files for enhancing the manuscript's quality and clarity.

I lack native proficiency and the necessary qualifications to evaluate the English standard of this manuscript.

Author Response

Reviewer 2

Dear reviewer 2, First of all, we would like to thank you for giving a timely review report/feedback.   The comments you have given are really very effective and comprehensive. We are grateful to you for helping us to improve our manuscript.  Below we provide detailed point to point responses to all comments and changes made. We hope our revisions are satisfactory and our paper can be accepted for publication.

I find the article intriguing as it presents promising prospects for advancements in this area through innovative solutions that can be applied to real-world situations. The article exhibits acceptable structure and well-written content. However, there are several crucial aspects that have either been inadequately explained or are noticeably absent from the text. Certain aspects have been minimized or left unexplained. Please find my suggestions my comments in the attached files for enhancing the manuscript's quality and clarity.

Reply: Thank you for giving us valuable comments and suggestions to improve the quality of the research content of the manuscript on. Our sincere thanks to the reviewer for his time and efforts. As per the reviewer’s comments, we have made the following corrections such as follows, the modified content of the manuscript highlighted in “red colour text” according to the reviewer’s comments.

1-A case study for evaluates the Index-Based Methods for Impervious Surface Mapping from Landsat-8 to Cities in Dry Climates; A Case Study of Buraydah City, KSA is presented in the manuscript. The content of the paper is informative; however, there are several sections that need to be revised and rewritten using proper organization. It is absolutely necessary for the authors to take into consideration revising the organization and composition of the manuscript in terms of the definition and justification of the objectives, the description of the method, the accomplishment of the objective, and the results.

Reply: The authors are also grateful to anonymous reviewers and the editor for their constructive comments which improved the manuscript. As suggested by the reviewer, necessary modifications and corrections are made in the revised manuscript

2-I recommend including four paragraphs in the introduction to describe the concept, the research gap, the contribution, and the paper's structure. The motivation has the potential to be expanded upon. You may include information regarding the necessity of conducting this research, the contribution of this article to the existing body of knowledge, etc. The discussion of the paper's originality is inadequate and it should be stated explicitly. In addition to the research question, the introduction must include a few words on the testable hypothesis. Please explain the significance of this work. Please discuss whether the paper has broad international applicability and interest.

Reply: As per reviewer suggested, all suggestions research gap, the contribution, the paper's structure, hypothesis and significance of this work   has been mentioned in introduction.

3-The abbreviated word should be thoroughly explained upon its first mention in the paper.

Reply: As per reviewer suggested, in the entire manuscript abbreviations have been defined first.

4-Support this sentence with proper references such as https://doi.org/10.3390/rs12081302 , 10.1007/978-3-030-21344-2_9.

Reply: Thank you for your feedback. We appreciate your suggestion to include more recent papers relevant to our study in the references. As per reviewer suggested, appropriate references have been attached.

5-Support this sentence with proper references such as  https://doi.org/10.3390/rs15041102

https://doi.org/10.46717/igj.55.2C.10ms-2022-08-23

Reply: As per reviewer suggested, appropriate references have been attached

6-Support this sentence with proper references such as

10.24425/jwld.2021.138166

https://doi.org/10.1016/j.jhydrol.2021.125974

Reply: As per reviewer suggested, appropriate references have been attached.

7-Kindly review the abbreviations thoroughly as I have come across numerous mistakes.

Reply: As per reviewer suggested some mistakes have been corrected.

8-The abbreviated word should be thoroughly explained upon its first mention in the paper

Reply: As per reviewer suggested, in the entire manuscript abbreviations have been defined first.

9-The authors failed to provide sufficient details about the novelty of this work, leaving room to enhance its value as a valuable contribution.

Reply: This study marks the first attempt to assess the performances of different spectral in- dices for ISA extraction in arid climate using two seasons Landsat-8 images.

10-I would suggest creating a clear and comprehensive flowchart to illustrate the methodology in a more understandable manner.

Reply: As per reviewer suggested, flowchart to illustrate the methodology have been attached

12- Km2

Reply: As per reviewer suggested some mistakes have been corrected.

13-610 m  Above Sea Level (A.S.L)

Reply: As per reviewer suggested some mistakes have been corrected.

14-some notes about the figure:

1It would be more clear for the reader if in the zoom out figure in the upper right to show whole Saudi Arabia with a neighboring country like Iraq, Oman, Jordan, Yemen as you know this will be for the international readers. I also recommend to show the administrative boundaries of the cities and show the capital city of Saudi Arabia.

2- At least its not clear for me what is the white lines inside the study area? I recommend to have the Legend of the Figure.

Reply: As per reviewer suggested, appropriate neu map have been added.

15-Why were satellite images from August 2020 acquired when the introduction section states that the ideal time for acquiring satellite images is during the spring season? There should be statement supporting this decision.

Reply: Thank you for pointing this out. Some studies have shown that the best time to extract ISA is the rainy season. In this study, we attempted to test the indices in a very dry period(summer) when there is no rainfall and in a relative dry period when there is some rainfall (20-30 mm) (spring). The goal is to find out which ISA Index are less affected by the changing climatic conditions in dry environment and to check the consistency of the results.

16- could not find this table? Where is it?

Reply: The table has been added.

17-Once a word is initially abbreviated within a sentence, there is no need to repeat the full word everywhere; instead, you can utilize the abbreviated form.

 Reply: We have ensured that the first instance of a word is now presented as an abbreviation throughout the manuscript.

18-What do you mean DT, which repeated few times?

Reply: The abbreviation has been corrected as TD

19-The abbreviated word should be thoroughly explained upon its first mention in the paper.

Reply: The abbreviation has been corrected as Otsu's method

20-Where can the legend of this Figure be found? How does one distinguish between colors or understand the meanings of red, gray and green?

Reply: The legend has been added.

21-Where can the legend of this Figure be found? How does one distinguish between colors or understand the meanings of red and yellow?

Reply: The legend has been added.

23-ASI and VgNIR

Reply: The abbreviations have been corrected as ISA and VrNIR-BI

24-Where is numbering?

Reply:The number has been added

25-The introduction is unnecessary; it is preferable to directly focus on discussing the results and outcomes of your study.

Reply: Most of the introduction in the "Discussion" section has been deleted.

26-The research question and hypothesis must be answered and discussed clearly in conclusion section. Please communicate the future research. The lessons learned must be further elaborated in the conclusion by discussing the results to the community and the future impacts. What is your perspective on future research?

Reply: As suggested by the reviewer, the conclusion section was remodified and added the main finding and future recommendations.

27-Please check reference no. 48

Reply: The reference has been corrected.

The authors are also grateful to anonymous reviewers and the editor for their constructive comments which improved the manuscript.
